# NODE DUPLICATION IMPROVES COLD-START LINK PREDICTION

## ABSTRACT

Graph Neural Networks (GNNs) are prominent in graph machine learning and have shown state-of-the-art performance in Link Prediction (LP) tasks. Nonetheless, recent studies show that GNNs struggle to produce good results on low-degree nodes despite their overall strong performance. In practical applications of LP, like recommendation systems, improving performance on low-degree nodes is critical, as it amounts to tackling the cold-start problem of improving the experiences of users with few observed interactions. In this paper, we investigate improving GNNs' LP performance on low-degree nodes while preserving their performance on high-degree nodes and propose a simple yet surprisingly effective augmentation technique called NODEDUP. Specifically, NODEDUP duplicates low-degree nodes and creates links between nodes and their own duplicates before following the standard supervised LP training scheme. By leveraging a "multi-view" perspective for low-degree nodes, NODEDUP shows significant LP performance improvements on low-degree nodes without compromising any performance on high-degree nodes. Additionally, as a plug-and-play augmentation module, NODEDUP can be easily applied on existing GNNs with very light computational cost. Extensive experiments show that NODEDUP achieves **38.49%**, **13.34%**, and **6.76%** relative improvements on isolated, low-degree, and warm nodes, respectively, on average across all datasets compared to GNNs and state-of-the-art cold-start methods.

## 1 INTRODUCTION

Link prediction (LP) is a fundamental task of graph-structured data (Liben-Nowell & Kleinberg, 2007; Trouillon et al., 2016), which aims to predict the likelihood of the links existing between two nodes in the network. It has wide-ranging real-world applications across different domains, such as friend recommendations in social media (Sankar et al., 2021; Tang et al., 2022; Fan et al., 2022), product recommendations in e-commerce platforms (Ying et al., 2018; He et al., 2020), knowledge graph completion (Li et al., 2023; Vashishth et al., 2020; Zhang et al., 2020), and chemical interaction prediction (Stanfield et al., 2017; Kovács et al., 2019; Yang et al., 2021).

In recent years, graph neural networks (GNNs) (Kipf & Welling, 2016a; Veličković et al., 2017; Hamilton et al., 2017) have been widely applied to LP, and a series of cutting-edge models have been proposed (Zhang & Chen, 2018; Zhang et al., 2021; Zhu et al., 2021; Zhao et al., 2022b). Most GNNs follow a message-passing scheme (Gilmer et al., 2017) in which information is iteratively aggregated from neighbors and used to update node representations accordingly. Consequently, the success of GNNs usually heavily relies on having sufficient high-quality neighbors for each node (Zheng et al., 2021; Liu et al., 2021). However, real-world graphs often exhibit long-tailed distribution in terms of node degrees, where a significant fraction of nodes have very few neighbors (Tang et al., 2020b; Ding et al., 2021; Hao et al., 2021). For example, Figure 1

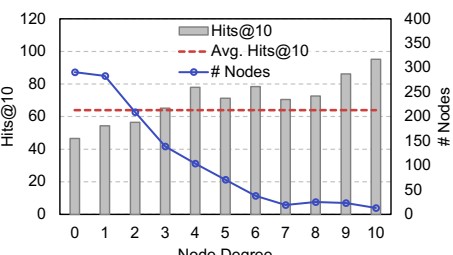

Figure 1: Node Degree Distribution and LP Performance (GSage as an encoder and inner product as a decoder) Distribution w.r.t Nodes Degrees showing reverse trends on `Citeseer` dataset.

shows the long-tailed degree distribution of the `Citeseer` dataset. Moreover, LP performances w.r.t. node degrees on this dataset also clearly indicate that GNNs struggle to generate satisfactory

results for nodes with low or zero degrees. For simplicity, in this paper, we refer to the nodes with low or zero degrees as *cold* nodes and the nodes with higher degrees as *warm* nodes.

To boost GNNs' performance on cold nodes, recent studies have proposed various training strategies (Liu et al., 2020; 2021; Zheng et al., 2021; Hu et al., 2022) and augmentation strategies (Hu et al., 2022; Rong et al., 2019; Zhao et al., 2022b) to improve representation learning quality. For instance, ColdBrew (Zheng et al., 2021) posits that training a powerful MLP can rediscover missing neighbor information for cold nodes; TailGNN (Liu et al., 2021) utilizes a cold-node-specific module to accomplish the same objective. However, such advanced training strategies (e.g., ColdBrew and TailGNN) share a notable drawback: they are trained with a bias towards cold nodes, which then sacrifices performance on warm nodes (empirically validated in Table 1). However, in real-world applications, both cold nodes and warm nodes are critical (Clauset et al., 2009). On the other hand, while augmentation methods such as LAGNN (Liu et al., 2022b) do not have such bias, they primarily focus on improving the overall performance of GNNs in LP tasks, which may be dominated by warm nodes due to their higher connectivity. Additionally, the augmentation methods usually introduce a significant amount of extra computational costs (empirically validated in Figure 5). In light of the existing work discussed above on improving LP performance for cold nodes, we are naturally motivated to explore the following crucial but rather unexplored research question:

***Can we improve LP performance on cold nodes without compromising warm node performance?***

We observe that cold node LP performance usually suffers because they are under-represented in standard supervised LP training due to their few (if any) connections. Given this observation, in this work, we introduce a simple yet effective augmentation method, NODEDUP, for improving LP performance on cold nodes. Specifically, NODEDUP duplicates cold nodes and establishes edges between each original cold node and its corresponding duplicate. Subsequently, we conduct standard supervised end-to-end training of GNNs on the augmented graph. To better understand why NODEDUP is able to improve LP performance for cold nodes, we thoroughly analyze it from multiple perspectives, during which we discover that this simple technique effectively offers a "multi-view" perspective of cold nodes during training. This "multi-view" perspective of the cold nodes acts similarly to an ensemble and drives performance improvements for these nodes. Additionally, our straightforward augmentation method provides valuable supervised training signals for cold nodes and especially isolated nodes. Furthermore, we also introduce NODEDUP(L), a lightweight variation of NODEDUP that adds only self-loop edges into training edges for cold nodes. NODEDUP(L) empirically offers up to a $1.3\times$ speedup over NODEDUP for the training process and achieves significant speedup over existing augmentation baselines. In our experiments, we comprehensively evaluate our method on seven benchmark datasets. Compared to GNNs and state-of-the-art cold-start methods, NODEDUP achieves **38.49%**, **13.34%**, and **6.76%** relative improvements on isolated, low-degree, and warm nodes, respectively, on average across all datasets. NODEDUP also greatly outperforms augmentation baselines on cold nodes, with comparable warm node performance. Finally, as plug-and-play augmentation methods, our methods are versatile and effective with different LP encoders/decoders. They also achieve significant performance in a more realistic inductive setting. Our code can be found at https://anonymous.4open.science/r/NodeDup-0241/README.md.

## 2 PRELIMINARIES

**Notation.** Let an attributed graph be $G = \{\mathcal{V}, \mathcal{E}, \mathbf{X}\}$, where $\mathcal{V}$ is the set of $N$ nodes and $\mathcal{E} \subseteq \mathcal{V} \times \mathcal{V}$ is the edges where each $e_{vu} \in \mathcal{E}$ indicates nodes $v$ and $u$ are linked. Let $\mathbf{X} \in \mathbb{R}^{N \times F}$ be the node attribute matrix, where $F$ is the attribute dimension. Let $\mathcal{N}_v$ be the set of neighbors of node $v$, i.e., $\mathcal{N}_v = \{u | e_{vu} \in \mathcal{E}\}$, and the degree of node $v$ is $|\mathcal{N}_v|$. We separate the set of nodes $\mathcal{V}$ into three disjoint sets $\mathcal{V}_{iso}$, $\mathcal{V}_{low}$, and $\mathcal{V}_{warm}$ by their degrees based on the threshold hyperparameter $\delta$[1]. For each node $v \in \mathcal{V}$, $v \in \mathcal{V}_{iso}$ if $|\mathcal{N}_v| = 0$; $v \in \mathcal{V}_{low}$ if $0 < |\mathcal{N}_v| \leq \delta$; $v \in \mathcal{V}_{warm}$ if $|\mathcal{N}_v| > \delta$. For ease of notation, we also use $\mathcal{V}_{cold} = \mathcal{V}_{iso} \cup \mathcal{V}_{low}$ to denote the cold nodes, which is the union of Isolated and Low-degree nodes.

**LP with GNNs.** In this work, we follow the commonly-used encoder-decoder framework for GNN-based LP (Kipf & Welling, 2016b; Berg et al., 2017; Schlichtkrull et al., 2018; Ying et al., 2018; Davidson et al., 2018; Zhu et al., 2021; Yun et al., 2021; Zhao et al., 2022b), where a GNN encoder learns the node representations and the decoder predicts the link existence probabilities given each

---

[1]This threshold $\delta$ is set as 2 in our experiments, based on observed performance gaps in LP on various datasets, as shown in Figure 1 and Figure 6. Further reasons for this threshold are detailed in Appendix D.1.

pair of node representations. Most GNNs follow the message passing design (Gilmer et al., 2017) that iteratively aggregate each node's neighbors' information to update its embeddings. Without the loss of generality, for each node $v$, the $l$-th layer of a GNN can be defined as

$$\boldsymbol{h}_v^{(l)} = \text{UPDATE}\big(\boldsymbol{h}_v^{(l-1)}, \boldsymbol{m}_v^{(l-1)}\big), \text{s.t. } \boldsymbol{m}_v^{(l-1)} = \text{AGG}\big(\{\boldsymbol{h}_u^{(l-1)}\} : \forall u \in \mathcal{N}_v\big), \quad (1)$$

where $\boldsymbol{h}_v^{(l)}$ is the $l$-th layer's output representation of node $v$, $\boldsymbol{h}_v^{(0)} = \boldsymbol{x_v}$, AGG($\cdot$) is the (typically permutation-invariant) aggregation function, and UPDATE($\cdot$) is the update function that combines node $v$'s neighbor embedding and its own embedding from the previous layer. For any node pair $v$ and $u$, the decoding process can be defined as $\hat{y}_{vu} = \sigma\big(\text{DECODER}(\boldsymbol{h}_v, \boldsymbol{h}_u)\big)$, where $\boldsymbol{h}_v$ is the GNN's output representation for node $v$ and $\sigma$ is the Sigmoid function. Following existing literature, we use inner product (Wang et al., 2021; Zheng et al., 2021) as the default DECODER.

The standard supervised LP training optimizes model parameters w.r.t. a training set, which is usually the union of all observed $M$ edges and $KM$ no-edge node pairs (as training with all $O(N^2)$ no-edges is infeasible in practice), where $K$ is the negative sampling rate ($K = 1$ usually). We use $\mathcal{Y} = \{0,1\}^{M+KM}$ to denote the training set labels, where $y_{vu} = 1$ if $e_{vu} \in \mathcal{E}$ and 0 otherwise.

**The Cold-Start Problem.** The cold-start problem is prevalent in various domains and scenarios. In recommendation systems (Chen et al., 2020; Lu et al., 2020; Hao et al., 2021; Zhu et al., 2019; Volkovs et al., 2017; Liu & Zheng, 2020), cold-start refers to the lack of sufficient interaction history for new users or items, which makes it challenging to provide accurate recommendations. Similarly, in the context of GNNs, the cold-start problem refers to performance in tasks involving cold nodes, which have few or no neighbors in the graph. As illustrated in Figure 1, GNNs usually struggle with cold nodes in LP tasks due to unreliable or missing neighbors' information. In this work, we focus on enhancing LP performance for cold nodes, specifically predicting the presence of links between a cold node $v \in \mathcal{V}_{cold}$ and target node $u \in \mathcal{V}$ (w.l.o.g.). Additionally, we aim to maintain satisfactory LP performance for warm nodes. Prior studies on cold-start problems (Tang et al., 2020b; Liu et al., 2021; Zheng et al., 2021) inspired this research direction.

## 3 NODE DUPLICATION TO IMPROVE COLD-START PERFORMANCE

We make a simple but powerful observation: *cold nodes are strongly under-represented in the LP training.* Given that they have few or even no directly connected neighbors, they hardly participate in the standard supervised LP training as described in Section 2. For example, a model will not see an isolated node unless it is randomly sampled as a negative training edge for another node. In light of such observations, our proposed augmentation technique is simple: we duplicate under-represented cold nodes. By both training and aggregating with the edges connecting the cold nodes with their duplications, cold nodes are able to gain better visibility in the training process, which allows the GNN-based LP models to learn better representations. In this section, we introduce NODEDUP in detail, followed by comprehensive analyses of why it works from different perspectives.

### 3.1 PROPOSED METHOD

The implementation of NODEDUP can be summarized into four simple steps: (1): duplicate all cold nodes to generate the augmented node set $\mathcal{V}' = \mathcal{V} \cup \mathcal{V}_{cold}$, whose node feature matrix is then $\mathbf{X}' \in \mathbb{R}^{(N+|\mathcal{V}_{cold}|) \times F}$. (2): for each cold node $v \in \mathcal{V}_{cold}$ and its duplication $v'$, add an edge between them and get the augmented edge set $\mathcal{E}' = \mathcal{E} \cup \{e_{vv'} : \forall v \in \mathcal{V}_{cold}\}$. (3): include the augmented edges into the training set and get $\mathcal{Y}' = \mathcal{Y} \cup \{y_{vv'} = 1 : \forall v \in \mathcal{V}_{cold}\}$. (4): proceed with the standard supervised LP training on the augmented graph $G' = \{\mathcal{V}', \mathcal{E}', \mathbf{X}'\}$ with augmented training set $\mathcal{Y}'$. We also summarize this whole process of NODEDUP in Algorithm 1 in Appendix C. The effects of duplication nodes and frequency are discussed in Appendix D.3.

**Time Complexity.** We discuss complexity of our method in terms of the training process on the augmented graph. We use GSage (Hamilton et al., 2017) and inner product decoder as the default architecture when demonstrating the following complexity (w.l.o.g). With the augmented graph, GSage has a complexity of $O(R^L(N + |\mathcal{V}_{cold}|)D^2)$, where $R$ represents the number of sampled neighbors for each node, $L$ is the number of GSage layers (Wu et al., 2020), and $D$ denotes the size of node representations. In comparison to the non-augmented graph, NODEDUP introduces an extra time complexity of $O(R^L|\mathcal{V}_{cold}|D^2)$. For the inner product decoder, we incorporate additionally $|\mathcal{V}_{cold}|$ positive edges and also sample $|\mathcal{V}_{cold}|$ negative edges into the training process, resulting in the extra time complexity of the decoder as $O((M + |\mathcal{V}_{cold}|)D)$. Given that all cold nodes have few

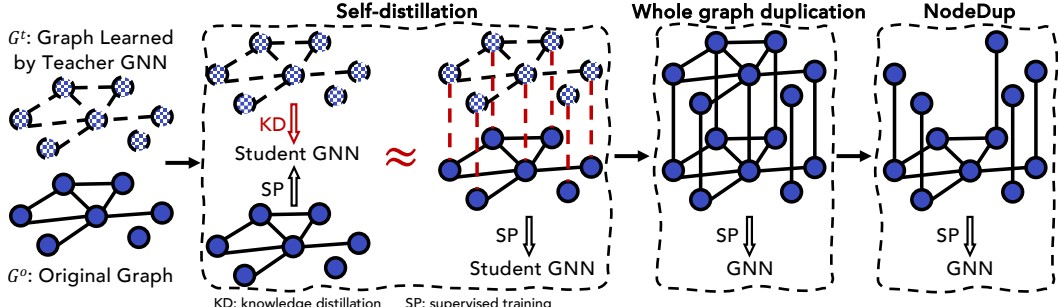

Figure 3: Comparing NODEDUP to self-distillation. The self-distillation process can be approximated by training the student GNN on an augmented graph, which combines $G^o$, $G^t$, and edges connecting corresponding nodes in the two graphs. This process can be further improved by replacing $G^t$ with $G^o$ to explore the whole graph duplication. NODEDUP is a lightweight variation of it.

($R \leq 2$ in our experiments) neighbors, and GSage is also always shallow (so $L$ is small) (Zhao & Akoglu, 2019), the overall extra complexity introduced by NODEDUP is $O(|\mathcal{V}_{cold}|D^2 + |\mathcal{V}_{cold}|D)$.

### 3.2 HOW DOES NODE DUPLICATION HELP COLD-START LP?

In this subsection, we analyze how such a simple method can improve cold-start LP from two perspectives: the neighborhood **aggregation** in GNNs and the **supervision** signal during training. In short, NODEDUP leverages the extra information from an additional "view". The existing view is when a node is regarded as the anchor node during message passing, whereas the additional view is when that node is regarded as one of its neighbors thanks to the duplicated node from NODEDUP.

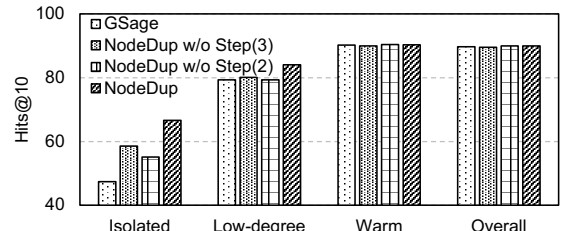

Figure 2: Ablation study of NODEDUP on Physics. Step (2) and Step (3) are the steps introduced in Section 3.1. Both steps play an important role in performance improvements of NODEDUP.

**Aggregation.** As described in Equation (1), when UPDATE($\cdot$) and AGG($\cdot$) do not share the transformation for node features, GNNs would have separate weights for self-representation and neighbor representations. The separate weights enable the neighbors and the node itself to play distinct roles in the UPDATE step. By leveraging this property, with NODEDUP, the model can leverage the two "views" for each node: first, the existing view is when a node is regarded as the anchor node during message passing, and the additional view is when that node is regarded as one of its neighbors thanks to the duplicated node from NODEDUP. Taking the official PyG (Fey & Lenssen, 2019) implementation of GSage (Hamilton et al., 2017) as an example, it updates node representations using $\boldsymbol{h}_v^{(l+1)} = W_1 \boldsymbol{h}_v^{(l)} + W_2 \boldsymbol{m}_v^{(l)}$. Here, $W_1$ and $W_2$ correspond to the self-representation and neighbors' representations, respectively. Without NODEDUP, isolated nodes $\mathcal{V}_{iso}$ have no neighbors, which results with $\boldsymbol{m}_v^{(l)} = \boldsymbol{0}$. Thus, the representations of all $v \in \mathcal{V}_{iso}$ are only updated by $\boldsymbol{h}_v^{(l+1)} = W_1 \boldsymbol{h}_v^{(l)}$. With NODEDUP, the updating process for isolated node $v$ becomes $\boldsymbol{h}_v^{(l+1)} = W_1 \boldsymbol{h}_v^{(l)} + W_2 \boldsymbol{h}_v^{(l)} = (W_1 + W_2)\boldsymbol{h}_v^{(l)}$. It indicates that $W_2$ is also incorporated into the node updating process for isolated nodes, which offers an additional perspective for isolated nodes' representation learning. Similarly, GAT (Veličković et al., 2017) updates node representations with $\boldsymbol{h}_v^{(l+1)} = \alpha_{vv}\Theta\boldsymbol{h}_v^{(l)} + \sum_{u \in \mathcal{N}_v} \alpha_{vu}\Theta\boldsymbol{h}_u^{(l)}$, where $\alpha_{vu} = \frac{\exp(\text{LeakyReLU}(\boldsymbol{a}^\top[\Theta\boldsymbol{h}_v^{(l)} || \Theta\boldsymbol{h}_u^{(l)}]))}{\sum_{i \in \mathcal{N}_v \cup v} \exp(\text{LeakyReLU}(\boldsymbol{a}^\top[\Theta\boldsymbol{h}_v^{(l)} || \Theta\boldsymbol{h}_i^{(l)}]))}$. Attention scores in $\boldsymbol{a}$ partially correspond to the self-representation $\boldsymbol{h}_v$ and partially to neighbors' representation $\boldsymbol{h}_u$. In this case, neighbor information offers a different perspective compared to self-representation. Such "multi-view" enriches the representations learned for the isolated nodes in a similar way to how ensemble methods work (Allen-Zhu & Li, 2020). Apart from addressing isolated nodes, the same mechanism and multi-view perspective also apply to Low-degree nodes.

**Supervision.** For LP tasks, besides the aggregation, edges also serve as supervised training signals. Cold nodes have few or no positive training edges connecting to them, potentially leading to out-of-distribution (OOD) issues (Wu et al., 2022), especially for isolated nodes. The additional edges, added

by NODEDUP to connect cold nodes with their duplicates, serve as additional positive supervision signals for LP. More supervision signals for cold nodes usually lead to better-quality embeddings.

**Ablation Study.** Figure 2 shows an ablation study on these two designs where NODEDUP w/o Step (3) indicates only using the augmented nodes and edges in aggregation but not supervision; NODEDUP w/o Step (2) indicates only using the augmented edges in supervision but not aggregation. We can observe that using augmented nodes/edges either in supervision or aggregation can significantly improve the LP performance on Isolated nodes, and NODEDUP, by combining them, results in larger improvements. Besides, NODEDUP also achieves improvements on Low-degree nodes while not sacrificing Warm nodes' performance.

**LP on Warm Nodes.** The superior performance on warm nodes is directly tied to our focus on link prediction tasks. Given the substantial number of Warm-Cold node pairs under prediction, these outcomes contribute to the overall performance metrics for both Warm node prediction. Better learning of Cold nodes thus boosts Cold-Warm node pairs link prediction performance, which subsequently elevates the prediction accuracy for warm nodes. A more detailed experimental analysis is provided in Appendix D.2.

### 3.3 RELATION BETWEEN NODEDUP AND SELF-DISTILLATION

Recently, Allen-Zhu & Li (2020) showed that the success of self-distillation, similar to our method, contributes to ensemble learning by providing models with different perspectives on the knowledge. Building on this insight, we show an interesting interpretation of NODEDUP, as a simplified and enhanced version of self-distillation for LP tasks for cold nodes, illustrated in Figure 3, in which we draw a connection between self-distillation and NODEDUP.

In **self-distillation**, a teacher GNN is first trained to learn the node representations $\mathbf{H}^t$ from original features $\mathbf{X}$. We denote the original graph as $G^o$, and we denote the graph, where we replace the node features in $G^o$ with $\mathbf{H}^t$, as $G^t$ in Figure 3. The student GNN is then initialized with random parameters and trained with the sum of two loss functions: $\mathcal{L}_{SD} = \mathcal{L}_{SP} + \mathcal{L}_{KD}$, where $\mathcal{L}_{SP}$ denotes the supervised training loss with $G^o$ and $\mathcal{L}_{KD}$ denotes the knowledge distillation loss with $G^t$. Figure 4 shows that self-distillation outperforms the teacher GNN across all settings.

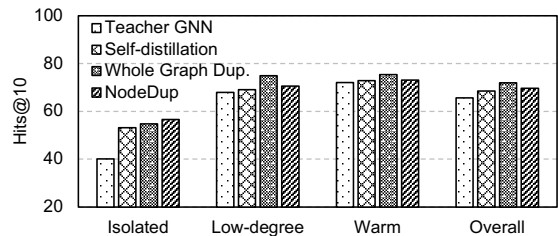

Figure 4: Performance with different training strategies introduced in Figure 3 on Citeseer. NODEDUP achieves better performance across all settings.

The effect of $\mathcal{L}_{KD}$ is similar to that of creating an additional link connecting nodes in $G^o$ to their corresponding nodes in $G^t$ when optimizing with $\mathcal{L}_{SP}$. This is illustrated by the red dashed line in Figure 3. For better clarity, we show the similarities between these two when we use the inner product as the decoder for LP with the following example. Given a node $v$ with normalized teacher embedding $\boldsymbol{h}_v^t$ and normalized student embedding $\boldsymbol{h}_v$, the additional loss term that would be added for distillation with cosine similarity is $\mathcal{L}_{KD} = -\frac{1}{N} \sum_{v \in \mathcal{V}} \boldsymbol{h}_v \cdot \boldsymbol{h}_v^t$. On the other hand, for the dashed line edges in Figure 3, we add an edge between the node $v$ and its corresponding node $v'$ in $G^t$ with embedding $\boldsymbol{h}_{v'}^t$. When trained with an inner product decoder and binary cross-entropy loss, it results in the following: $\mathcal{L}_{SP} = -\frac{1}{N} \sum y_{vv'} \log(\boldsymbol{h}_v \cdot \boldsymbol{h}_{v'}^t) + (1 - y_{vv'}) \log(1 - \boldsymbol{h}_v \cdot \boldsymbol{h}_{v'}^t)$. Since we always add the edge $(v, v')$, we know $y_{vv'} = 1$, and can simplify the loss as follows: $\mathcal{L}_{SP} = -\frac{1}{N} \sum \log(\boldsymbol{h}_v \cdot \boldsymbol{h}_{v'}^t)$. Here, we can observe that $\mathcal{L}_{KD}$ and $\mathcal{L}_{SP}$ are positively correlated as $\log(\cdot)$ is a monotonically increasing function.

To further improve this step and mitigate potential noise in $G^t$, we explore a whole graph duplication technique, where $G^t$ is replaced with an exact duplicate of $G^o$ to train the student GNN. The results in Figure 4 demonstrate significant performance enhancement achieved by **whole graph duplication** compared to self-distillation. NODEDUP is a lightweight variation of the whole graph duplication technique, which focuses on duplicating only the cold nodes and adding edges connecting them to their duplicates. From the results, it is evident that **NODEDUP** consistently outperforms the teacher GNN and self-distillation in all scenarios. Additionally, NODEDUP exhibits superior performance on isolated nodes and is much more efficient compared to the whole graph duplication approach.

### 3.4 NODEDUP(L): AN EFFICIENT VARIANT OF NODEDUP

Inspired by the above analysis, we further introduce a lightweight variant of NODEDUP for better efficiency, NODEDUP(L). To provide above-described "multi-view" information as well as the supervision signals for cold nodes, NODEDUP(L) simply add additional self-loop edges for the cold nodes into the edge set $\mathcal{E}$, that is, $\mathcal{E}' = \mathcal{E} \cup \{e_{vv} : \forall v \in \mathcal{V}_{cold}\}$. NODEDUP(L) preserves the two essential designs of NODEDUP while avoiding the addition of extra nodes, which further saves time and space complexity. Moreover, NODEDUP differs from NODEDUP(L) since each duplicated node in NODEDUP will provide another view for itself because of dropout layers, which leads to different performance as shown in Section 4.2.

**NODEDUP(L) vs. Self-loop.** We remark upon a resemblance between NODEDUP(L) and self-loops in GNNs (e.g., the additional self-connection in the normalized adjacency matrix by GCN) as they both add self-loop edges. However, they differ in two aspects. During **aggregation**: NODEDUP(L) intentionally incorporates the self-representation $\boldsymbol{h}_v^{(l)}$ into the aggregated neighbors' representation $\boldsymbol{m}_v^{(l)}$ by adding additional edges. Taking GSage as an example, the weight matrix $W_2$ would serve an extra "view" of $\boldsymbol{h}_v^{(l)}$ when updating $\boldsymbol{h}_v^{(l+1)}$, whereas the default self-loops only use information from $W_1$. Additionally, in the **supervision** signal: unlike the normal self-loops and the self-loops introduced in previous works (Cai et al., 2019; Wang et al., 2020), where self-loops are solely for aggregation, the edges added by NODEDUP(L) also serve as positive training samples for cold nodes.

## 4 EXPERIMENTS

### 4.1 EXPERIMENTAL SETTINGS

**Datasets and Evaluation Settings.** We conduct experiments on 7 benchmark datasets: `Cora`, `Citeseer`, `CS`, `Physics`, `Computers`, `Photos` and `IGB-100K`, with their details specified in Appendix B. We randomly split edges into training, validation, and testing sets. We allocated 10% for validation and 40% for testing in `Computers` and `Photos`, 5%/10% for testing in `IGB-100K`, and 10%/20% in other datasets. We follow the standard evaluation metrics used in the Open Graph Benchmark (Hu et al., 2020) for LP, in which we rank missing references higher than 500 negative reference candidates for each node. The negative references are randomly sampled from nodes not connected to the source node. We use Hits@10 as the main evaluation metric (Han et al., 2022) and also report MRR performance in Appendix D. We follow Guo et al. (2022) and Shiao et al. (2022) for the inductive settings, where new nodes appear after the training process. Additionally, results for large-scale datasets and heterophilic graphs are presented in Appendix D.4 and Appendix D.5.

**Baselines.** Both NODEDUP and NODEDUP(L) are flexible to integrate with different GNN encoder architectures and LP decoders. For our experiments, we use GSage (Hamilton et al., 2017) encoder and the inner product decoder as the default base LP model. To comprehensively evaluate our work, we compare NODEDUP against three categories of baselines. (1) Base LP models. (2) Cold-start methods: TailGNN (Liu et al., 2021) and Cold-brew (Zheng et al., 2021) primarily aim to enhance the performance on cold nodes. We also compared with Imbalance (Lin et al., 2017), viewing cold nodes as an issue of the imbalance concerning node degrees. (3) Graph data augmentation methods: Augmentation frameworks including DropEdge (Rong et al., 2019), TuneUP (Hu et al., 2022), and LAGNN (Liu et al., 2022b) typically improve the performance while introducing additional preprocessing or training time. Performance comparisons with heuristic methods are in Appendix D.6.

### 4.2 PERFORMANCE COMPARED TO BASE GNN LP MODELS

**Isolated and Low-degree Nodes.** We compare our methods with base GNN LP models that consist of a GNN encoder in conjunction with an inner product decoder and are trained with a supervised loss. From Table 1, we observe consistent improvements for both NODEDUP(L) and NODEDUP over the base GSage model across all datasets, particularly in the Isolated and Low-degree node settings. Notably, in the Isolated setting, NODEDUP achieves an impressive 29.6% improvement, on average, across all datasets. These findings provide clear evidence that our methods effectively address the issue of sub-optimal LP performance on cold nodes.

**Warm Nodes and Overall.** It is encouraging to see that NODEDUP(L) consistently outperforms GSage across all the datasets in the Warm nodes and Overall settings. NODEDUP also outperforms GSage in 13 out of 14 cases under both settings. These findings support the notion that our methods can effectively maintain and enhance the performance of Warm nodes.

Table 1: Performance compared with base GNN model and baselines for cold-start methods(evaluated by Hits@10). The best result is **bold**, and the runner-up is underlined. NODEDUP and NODEDUP(L) outperform GSage and cold-start baselines almost all the cases.

| | | GSage | Imbalance | TailGNN | Cold-brew | NODEDUP(L) | NODEDUP |
|---|---|---|---|---|---|---|---|
| Cora | Isolated | $32.20_{\pm3.58}$ | $34.51_{\pm1.11}$ | $36.95_{\pm1.34}$ | $28.17_{\pm0.67}$ | $\underline{39.76}_{\pm1.32}$ | $\mathbf{44.27}_{\pm3.82}$ |
| | Low-degree | $59.45_{\pm1.09}$ | $59.42_{\pm1.21}$ | $61.35_{\pm0.79}$ | $57.27_{\pm0.63}$ | $\mathbf{62.53}_{\pm1.03}$ | $\underline{61.98}_{\pm1.14}$ |
| | Warm | $\underline{61.14}_{\pm0.78}$ | $59.54_{\pm0.46}$ | $60.61_{\pm0.90}$ | $56.28_{\pm0.81}$ | $\mathbf{62.07}_{\pm0.37}$ | $59.07_{\pm0.68}$ |
| | Overall | $58.31_{\pm0.68}$ | $57.55_{\pm0.67}$ | $\underline{59.02}_{\pm0.71}$ | $54.44_{\pm0.53}$ | $\mathbf{60.49}_{\pm0.49}$ | $58.92_{\pm0.82}$ |
| Citeseer | Isolated | $47.13_{\pm2.43}$ | $46.26_{\pm0.86}$ | $37.84_{\pm3.36}$ | $37.78_{\pm4.23}$ | $\underline{52.46}_{\pm1.16}$ | $\mathbf{57.54}_{\pm1.04}$ |
| | Low-degree | $61.88_{\pm0.79}$ | $61.90_{\pm0.60}$ | $62.06_{\pm1.73}$ | $59.12_{\pm9.97}$ | $\underline{73.71}_{\pm1.22}$ | $\mathbf{75.50}_{\pm0.39}$ |
| | Warm | $71.45_{\pm0.52}$ | $71.54_{\pm0.86}$ | $71.32_{\pm1.83}$ | $65.12_{\pm7.82}$ | $\mathbf{74.99}_{\pm0.37}$ | $\underline{74.68}_{\pm0.67}$ |
| | Overall | $63.77_{\pm0.83}$ | $63.66_{\pm0.43}$ | $62.02_{\pm1.89}$ | $58.03_{\pm7.72}$ | $\underline{70.34}_{\pm0.35}$ | $\mathbf{71.73}_{\pm0.47}$ |
| CS | Isolated | $56.41_{\pm1.61}$ | $46.60_{\pm1.66}$ | $55.70_{\pm1.38}$ | $57.70_{\pm0.81}$ | $\underline{65.18}_{\pm1.25}$ | $\mathbf{65.87}_{\pm1.70}$ |
| | Low-degree | $75.95_{\pm0.25}$ | $75.53_{\pm0.21}$ | $73.60_{\pm0.70}$ | $73.99_{\pm0.34}$ | $\mathbf{81.46}_{\pm0.57}$ | $\underline{81.12}_{\pm0.36}$ |
| | Warm | $84.37_{\pm0.46}$ | $83.70_{\pm0.46}$ | $79.86_{\pm0.35}$ | $78.23_{\pm0.28}$ | $\mathbf{85.48}_{\pm0.26}$ | $\underline{84.76}_{\pm0.41}$ |
| | Overall | $83.33_{\pm0.42}$ | $82.56_{\pm0.40}$ | $79.05_{\pm0.36}$ | $77.63_{\pm0.23}$ | $\mathbf{84.90}_{\pm0.29}$ | $\underline{84.23}_{\pm0.39}$ |
| Physics | Isolated | $47.41_{\pm1.38}$ | $55.01_{\pm0.58}$ | $52.54_{\pm1.34}$ | $64.38_{\pm0.85}$ | $\underline{65.04}_{\pm0.63}$ | $\mathbf{66.65}_{\pm0.95}$ |
| | Low-degree | $79.31_{\pm0.28}$ | $79.50_{\pm0.27}$ | $75.95_{\pm0.27}$ | $75.86_{\pm0.10}$ | $\underline{82.70}_{\pm0.22}$ | $\mathbf{84.04}_{\pm0.22}$ |
| | Warm | $90.28_{\pm0.23}$ | $89.85_{\pm0.09}$ | $85.93_{\pm0.40}$ | $78.48_{\pm0.14}$ | $\mathbf{90.44}_{\pm0.23}$ | $\underline{90.33}_{\pm0.05}$ |
| | Overall | $89.76_{\pm0.22}$ | $89.38_{\pm0.09}$ | $85.48_{\pm0.38}$ | $78.34_{\pm0.13}$ | $\mathbf{90.09}_{\pm0.22}$ | $\underline{90.03}_{\pm0.05}$ |
| Computers | Isolated | $9.32_{\pm1.44}$ | $10.14_{\pm0.59}$ | $10.63_{\pm1.59}$ | $9.75_{\pm1.24}$ | $\underline{17.11}_{\pm1.62}$ | $\mathbf{19.62}_{\pm2.63}$ |
| | Low-degree | $57.91_{\pm0.97}$ | $56.19_{\pm0.82}$ | $51.21_{\pm1.58}$ | $49.03_{\pm0.94}$ | $\mathbf{62.14}_{\pm1.06}$ | $\underline{61.16}_{\pm0.92}$ |
| | Warm | $66.87_{\pm0.47}$ | $65.62_{\pm0.21}$ | $62.77_{\pm0.44}$ | $57.52_{\pm0.28}$ | $\underline{68.02}_{\pm0.41}$ | $\mathbf{68.10}_{\pm0.25}$ |
| | Overall | $66.67_{\pm0.47}$ | $65.42_{\pm0.20}$ | $62.55_{\pm0.45}$ | $57.35_{\pm0.28}$ | $\underline{67.86}_{\pm0.41}$ | $\mathbf{67.94}_{\pm0.25}$ |
| Photos | Isolated | $9.25_{\pm2.31}$ | $10.80_{\pm1.72}$ | $13.62_{\pm1.00}$ | $12.86_{\pm2.58}$ | $\mathbf{21.50}_{\pm2.14}$ | $\underline{17.84}_{\pm3.53}$ |
| | Low-degree | $52.61_{\pm0.88}$ | $50.68_{\pm0.57}$ | $42.75_{\pm2.50}$ | $43.14_{\pm0.64}$ | $\mathbf{55.70}_{\pm1.38}$ | $\underline{54.13}_{\pm1.58}$ |
| | Warm | $67.64_{\pm0.55}$ | $64.54_{\pm0.50}$ | $61.63_{\pm0.73}$ | $58.06_{\pm0.56}$ | $\mathbf{69.68}_{\pm0.87}$ | $\underline{68.68}_{\pm0.49}$ |
| | Overall | $67.32_{\pm0.54}$ | $64.24_{\pm0.49}$ | $61.29_{\pm0.75}$ | $57.77_{\pm0.56}$ | $\mathbf{69.40}_{\pm0.86}$ | $\underline{68.39}_{\pm0.48}$ |
| IGB-100K | Isolated | $75.92_{\pm0.52}$ | $77.32_{\pm0.79}$ | $77.29_{\pm0.34}$ | $82.31_{\pm0.30}$ | $\underline{87.43}_{\pm0.44}$ | $\mathbf{88.04}_{\pm0.20}$ |
| | Low-degree | $79.38_{\pm0.23}$ | $79.19_{\pm0.09}$ | $80.57_{\pm0.14}$ | $83.84_{\pm0.16}$ | $\underline{88.37}_{\pm0.24}$ | $\mathbf{88.98}_{\pm0.17}$ |
| | Warm | $86.42_{\pm0.24}$ | $86.01_{\pm0.19}$ | $85.35_{\pm0.19}$ | $82.44_{\pm0.21}$ | $\mathbf{88.54}_{\pm0.31}$ | $\underline{88.28}_{\pm0.20}$ |
| | Overall | $84.77_{\pm0.21}$ | $84.47_{\pm0.14}$ | $84.19_{\pm0.18}$ | $82.68_{\pm0.17}$ | $\mathbf{88.47}_{\pm0.28}$ | $\underline{88.39}_{\pm0.18}$ |

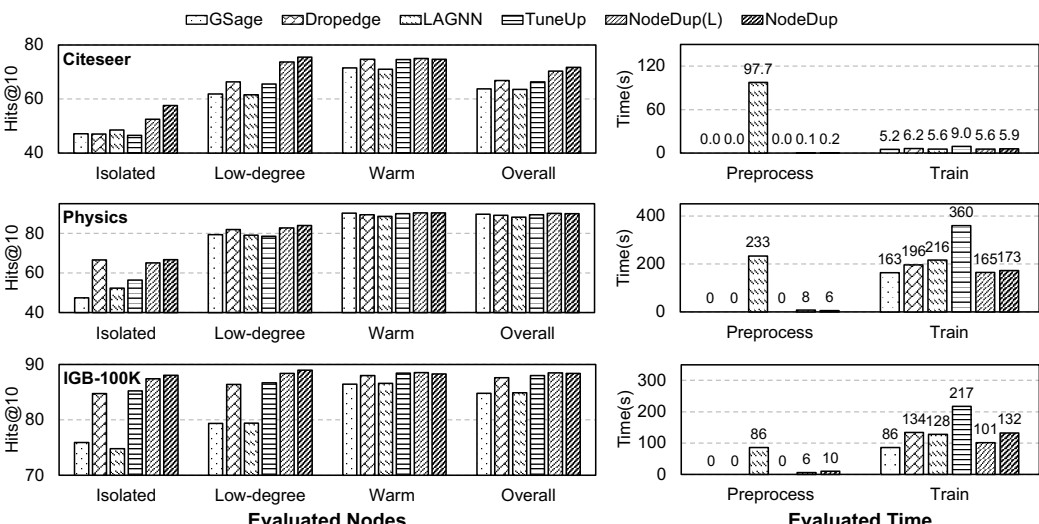

Figure 5: Performance and runtime comparisons of different augmentation methods. The *left* histograms show the performance results, and the *right* histograms show the preprocessing and training time consumption of each method. Our methods consistently achieve significant improvements in both performance for Isolated and Low-degree node settings and runtime efficiency over baselines.

**NODEDUP vs. NODEDUP(L).** Furthermore, we observe that NODEDUP achieves greater improvements over NODEDUP(L) for Isolated nodes. However, NODEDUP(L) outperforms NODEDUP on 6 out of 7 datasets for Warm nodes. The additional improvements achieved by NODEDUP for Isolated nodes can be attributed to the extra view provided to cold nodes through node duplication during aggregation. On the other hand, the impact of node duplication on the original graph structure likely

affects the performance of Warm nodes, which explains the superior performance of NODEDUP(L) in this setting compared to NODEDUP.

### 4.3 PERFORMANCE COMPARE TO COLD-START METHODS

Table 1 presents the LP performance of various cold-start baselines. For both Isolated and Low-degree nodes, we consistently observe *substantial improvements* of our NODEDUP and NODEDUP(L) methods compared to other cold-start baselines. Specifically, NODEDUP and NODEDUP(L) achieve 38.49% and 34.74% improvement for Isolated nodes on average across all datasets, respectively.

In addition, our methods consistently outperform cold-start baselines for Warm nodes across all the datasets, where NODEDUP(L) and NODEDUP achieve 6.76% and 7.95% improvements on average, respectively. This shows that our methods can successfully overcome issues with degrading performance on Warm nodes in cold-start baselines. Further analyses with other cold-start methods and efficiency comparisons can be found in Appendix D.8 and Appendix D.9.

### 4.4 PERFORMANCE COMPARED TO AUGMENTATION METHODS

**Effectiveness Comparison.** Since NODEDUP and NODEDUP(L) use graph data augmentation techniques, we compare them to other data augmentation baselines. The performance and time consumption results are presented in Figure 5 for three datasets (Citeseer, Physics, and IGB-100K), while the results for the remaining datasets are provided in Appendix D.10 due to the page limit. From Figure 5, we consistently observe that NODEDUP outperforms all the graph augmentation baselines for Isolated and Low-degree nodes across all three datasets. Similarly, NODEDUP(L) outperforms graph data augmentation baselines on 17/18 cases for Isolated and Low-degree nodes. Not only did our methods perform better for Isolated and Low-degree nodes, NODEDUP and NODEDUP(L) also perform on par or above baselines for Warm nodes.

**Efficiency Comparison.** Augmentation methods often come with the trade-off of adding additional run time before or during model training. For example, LAGNN (Liu et al., 2022b) requires extra preprocessing time to train the generative model prior to GNN training. It also takes additional time to generate extra features for each node during training. Although Dropedge (Rong et al., 2019) and TuneUP (Hu et al., 2022) are free of preprocessing, they require additional time to drop edges in each training epoch compared to base GNN training. Furthermore, the two-stage training employed by TuneUP doubles the training time compared to one-stage training methods. For NODEDUP methods, duplicating nodes and adding edges is remarkably swift and consumes significantly less preprocessing time than other augmentation methods. As an example, NODEDUP(L) and NODEDUP are **977.0×** and **488.5×** faster than LAGNN in preprocessing Citeseer, respectively. We also observe that NODEDUP(L) has the least training time among all augmentation methods and datasets, while NODEDUP also requires less training time in 8/9 cases. Additionally, NODEDUP(L) achieves significant efficiency benefits compared to NODEDUP in Figure 5, especially when the number of nodes in the graph increases substantially. Taking the IGB-100K dataset as an example, NODEDUP(L) is 1.3× faster than NODEDUP for the entire training process.

### 4.5 PERFORMANCE UNDER THE INDUCTIVE SETTING

Under the inductive setting (Guo et al., 2022; Shiao et al., 2022), which closely resembles real-world LP scenarios, the presence of new nodes after the training stage adds an additional challenge compared to the transductive setting. We evaluate and present the effectiveness of our methods under this setting in Table 2 for Citeseer, Physics, and IGB-100K datasets. Additional results for other datasets can be found in Appendix D.11. In Table 2, we observe that our methods consistently outperform base GSage across all of the datasets. We also observe significant performance im-

Table 2: Performance in inductive settings (evaluated by Hits@10). The best result is **bold**, and the runner-up is underlined. Our methods consistently outperform GSage.

|  |  | GSage | NODEDUP(L) | NODEDUP |
|---|---|---|---|---|
| Citeseer | Isolated | $58.42_{\pm0.49}$ | $\underline{62.42}_{\pm1.88}$ | $\mathbf{62.94}_{\pm1.91}$ |
|  | Low-degree | $67.75_{\pm1.06}$ | $\underline{69.93}_{\pm1.18}$ | $\mathbf{72.05}_{\pm1.23}$ |
|  | Warm | $72.98_{\pm1.15}$ | $\mathbf{75.04}_{\pm1.03}$ | $\underline{74.40}_{\pm2.43}$ |
|  | Overall | $66.98_{\pm0.61}$ | $\underline{69.65}_{\pm0.83}$ | $\mathbf{70.26}_{\pm1.16}$ |
| Physics | Isolated | $85.62_{\pm0.23}$ | $\underline{85.94}_{\pm0.15}$ | $\mathbf{86.90}_{\pm0.35}$ |
|  | Low-degree | $80.87_{\pm0.43}$ | $\underline{81.23}_{\pm0.56}$ | $\mathbf{85.56}_{\pm0.25}$ |
|  | Warm | $90.22_{\pm0.36}$ | $\underline{90.37}_{\pm0.25}$ | $\mathbf{90.54}_{\pm0.14}$ |
|  | Overall | $89.40_{\pm0.33}$ | $\underline{89.57}_{\pm0.23}$ | $\mathbf{89.98}_{\pm0.13}$ |
| IGB-100K | Isolated | $84.33_{\pm0.87}$ | $\underline{92.94}_{\pm0.11}$ | $\mathbf{93.95}_{\pm0.06}$ |
|  | Low-degree | $93.19_{\pm0.06}$ | $\underline{93.33}_{\pm0.11}$ | $\mathbf{94.00}_{\pm0.09}$ |
|  | Warm | $90.76_{\pm0.13}$ | $\mathbf{91.21}_{\pm0.07}$ | $\underline{91.20}_{\pm0.08}$ |
|  | Overall | $90.31_{\pm0.18}$ | $\underline{91.92}_{\pm0.05}$ | $\mathbf{92.21}_{\pm0.04}$ |

Table 3: Performance with different encoders (inner product as the decoder). The best result for each encoder is **bold**, and the runner-up is underlined. Our methods consistently outperform the base models, particularly for Isolated and Low-degree nodes.

| | | GAT | NODEDUP(L) | NODEDUP | JKNet | NODEDUP(L) | NODEDUP |
|---|---|---|---|---|---|---|---|
| Citeseer | Isolated | $37.78_{\pm 2.36}$ | $\underline{38.95}_{\pm 2.75}$ | $\mathbf{44.04}_{\pm 1.03}$ | $37.78_{\pm 0.63}$ | $\underline{49.06}_{\pm 0.60}$ | $\mathbf{55.15}_{\pm 0.87}$ |
| | Low-degree | $58.04_{\pm 2.40}$ | $\underline{61.93}_{\pm 1.66}$ | $\mathbf{66.73}_{\pm 0.96}$ | $60.74_{\pm 1.18}$ | $\underline{71.78}_{\pm 0.64}$ | $\mathbf{75.26}_{\pm 1.16}$ |
| | Warm | $56.37_{\pm 2.15}$ | $\underline{64.55}_{\pm 1.74}$ | $\mathbf{66.61}_{\pm 1.67}$ | $71.61_{\pm 0.76}$ | $\underline{74.66}_{\pm 0.47}$ | $\mathbf{75.81}_{\pm 0.89}$ |
| | Overall | $53.42_{\pm 1.59}$ | $\underline{58.89}_{\pm 0.89}$ | $\mathbf{62.41}_{\pm 0.78}$ | $61.73_{\pm 0.57}$ | $\underline{68.91}_{\pm 0.38}$ | $\mathbf{71.75}_{\pm 0.82}$ |
| Physics | Isolated | $38.19_{\pm 1.23}$ | $\underline{39.95}_{\pm 1.48}$ | $\mathbf{45.89}_{\pm 2.82}$ | $42.57_{\pm 1.93}$ | $\underline{55.47}_{\pm 2.25}$ | $\mathbf{61.11}_{\pm 2.27}$ |
| | Low-degree | $74.19_{\pm 0.31}$ | $\underline{74.77}_{\pm 0.46}$ | $\mathbf{76.36}_{\pm 0.25}$ | $75.36_{\pm 0.23}$ | $\underline{79.55}_{\pm 0.21}$ | $\mathbf{81.14}_{\pm 0.28}$ |
| | Warm | $85.84_{\pm 0.32}$ | $\mathbf{86.02}_{\pm 0.45}$ | $85.84_{\pm 0.15}$ | $88.24_{\pm 0.32}$ | $\mathbf{89.42}_{\pm 0.16}$ | $\underline{89.24}_{\pm 0.16}$ |
| | Overall | $\mathbf{85.27}_{\pm 0.30}$ | $\underline{85.47}_{\pm 0.45}$ | $85.37_{\pm 0.14}$ | $87.64_{\pm 0.31}$ | $\mathbf{88.96}_{\pm 0.15}$ | $\underline{88.87}_{\pm 0.15}$ |
| IGB-100K | Isolated | $75.87_{\pm 0.48}$ | $\underline{78.17}_{\pm 0.58}$ | $\mathbf{80.18}_{\pm 0.31}$ | $69.29_{\pm 0.73}$ | $\underline{86.60}_{\pm 0.46}$ | $\mathbf{86.85}_{\pm 0.41}$ |
| | Low-degree | $77.05_{\pm 0.15}$ | $\underline{78.50}_{\pm 0.31}$ | $\mathbf{81.00}_{\pm 0.12}$ | $76.90_{\pm 0.27}$ | $\underline{86.94}_{\pm 0.15}$ | $\mathbf{87.65}_{\pm 0.20}$ |
| | Warm | $81.40_{\pm 0.07}$ | $\mathbf{81.95}_{\pm 0.25}$ | $81.19_{\pm 0.20}$ | $84.93_{\pm 0.30}$ | $\mathbf{87.41}_{\pm 0.13}$ | $\underline{86.19}_{\pm 0.12}$ |
| | Overall | $80.42_{\pm 0.07}$ | $\mathbf{81.19}_{\pm 0.25}$ | $\underline{81.11}_{\pm 0.19}$ | $82.91_{\pm 0.28}$ | $\mathbf{87.29}_{\pm 0.13}$ | $\underline{86.47}_{\pm 0.13}$ |

provements of our methods on Isolated nodes, where NODEDUP and NODEDUP(L) achieve 5.50% and 3.57% improvements averaged across the three datasets, respectively. Additionally, NODEDUP achieves 5.09% improvements on Low-degree nodes. NODEDUP leads to more pronounced improvements on Low-degree/Isolated nodes, making it particularly beneficial for the inductive setting.

### 4.6 PERFORMANCE WITH DIFFERENT ENCODERS/DECODERS

As a simple plug-and-play augmentation method, NODEDUP can work with different GNN encoders and LP decoders. In Tables 3 and 4, we present results with GAT (Veličković et al., 2017) and JKNet (Xu et al., 2018) as encoders, along with a MLP decoder. Due to the space limit, we only report the results of three datasets here and leave the remaining in Appendix D.12. When applying NODEDUP to base LP training, with GAT or JKNet as the encoder and inner product as the decoder, we observe significant performance improvements across the board. Regardless of the encoder choice, NODEDUP consistently outperforms the base models, particularly for Isolated and Low-degree nodes. From Appendix D.12, we also observe the performance improvements of NODEDUP with GCN (Kipf & Welling, 2016a), GraphTransformer (Dwivedi & Bresson, 2020) as encoders.

In Table 4, we present the results of our methods applied to the base LP training, where GSage serves as the encoder and MLP as the decoder. Regardless of the decoder, we observe better performance with our methods. These improvements are significantly higher compared to the improvements observed with the inner product decoder. The primary reason for this discrepancy is the inclusion of additional supervised training signals for isolated nodes in our methods, as discussed in Section 3.2. These signals play a crucial role in training the MLP decoder,

Table 4: LP performance with MLP decoder (GSage as the encoder). Our methods outperform the base model.

| | | MLP-Dec. | NODEDUP(L) | NODEDUP |
|---|---|---|---|---|
| Citeseer | Isolated | $17.16_{\pm 1.14}$ | $\underline{37.84}_{\pm 3.06}$ | $\mathbf{51.17}_{\pm 2.19}$ |
| | Low-degree | $63.82_{\pm 1.58}$ | $\underline{68.49}_{\pm 1.19}$ | $\mathbf{71.98}_{\pm 1.29}$ |
| | Warm | $72.93_{\pm 1.25}$ | $\underline{75.33}_{\pm 0.54}$ | $\mathbf{75.72}_{\pm 0.55}$ |
| | Overall | $59.49_{\pm 1.21}$ | $\underline{66.07}_{\pm 0.74}$ | $\mathbf{69.89}_{\pm 0.65}$ |
| Physics | Isolated | $11.59_{\pm 1.88}$ | $\mathbf{60.25}_{\pm 2.54}$ | $\underline{59.50}_{\pm 1.87}$ |
| | Low-degree | $76.37_{\pm 0.64}$ | $\underline{81.74}_{\pm 0.77}$ | $\mathbf{82.58}_{\pm 0.79}$ |
| | Warm | $91.54_{\pm 0.33}$ | $\mathbf{91.96}_{\pm 0.36}$ | $\underline{91.59}_{\pm 0.22}$ |
| | Overall | $90.78_{\pm 0.33}$ | $\mathbf{91.51}_{\pm 0.38}$ | $\underline{91.13}_{\pm 0.23}$ |
| IGB-100K | Isolated | $3.51_{\pm 0.32}$ | $\mathbf{82.71}_{\pm 1.05}$ | $\underline{82.02}_{\pm 0.73}$ |
| | Low-degree | $75.25_{\pm 0.49}$ | $\underline{85.96}_{\pm 0.42}$ | $\mathbf{86.04}_{\pm 0.26}$ |
| | Warm | $85.06_{\pm 0.08}$ | $\mathbf{87.89}_{\pm 0.13}$ | $\underline{86.87}_{\pm 0.48}$ |
| | Overall | $80.16_{\pm 0.16}$ | $\mathbf{87.35}_{\pm 0.21}$ | $\underline{86.54}_{\pm 0.40}$ |

making it more responsive to the specific challenges presented by isolated nodes. Our methods also improve performance with SEAL (Zhang & Chen, 2018), as shown in Appendix D.12.

## 5 CONCLUSION

GNNs in LP encounter difficulties when dealing with cold nodes that lack sufficient or absent neighbors. To address this challenge, we presented a simple yet effective augmentation method (NODEDUP) specifically tailored for the cold-start LP problem, which can effectively enhance the prediction capabilities of GNNs for cold nodes while maintaining overall performance. Extensive evaluations demonstrated that both NODEDUP and its lightweight variant, NODEDUP(L), consistently outperformed baselines on both cold node and warm node settings across 7 benchmark datasets. NODEDUP also achieved better runtime efficiency compared to the augmentation baselines.

**Ethics Statement.** In this work, our simple but effective method enhances the link prediction performance on cold-start nodes, which mitigates the degree bias and advances the fairness of graph machine learning. It can be widely used and beneficial for various real-world applications, such as recommendation systems, social network analysis, and bioinformatics. We do not foresee any negative societal impact or ethical concerns posed by our method. Nonetheless, we note that both positive and negative societal impacts can be made by applications of graph machine learning techniques, which may benefit from the improvements induced by our work. Care must be taken, in general, to ensure positive societal and ethical consequences of machine learning.

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

# CONTENTS OF APPENDIX

## A  RELATED WORK

**LP with GNNs.** Over the past few years, GNN architectures (Kipf & Welling, 2016a; Gilmer et al., 2017; Hamilton et al., 2017; Veličković et al., 2017; Xu et al., 2018) have gained significant attention and demonstrated promising outcomes in LP tasks. There are two primary approaches to applying GNNs in LP. The first approach involves a node-wise encoder-decoder framework, which we discussed in Section 2. The second approach reformulates LP tasks as enclosing subgraph classification tasks (Zhang & Chen, 2018; Cai & Ji, 2020; Cai et al., 2021; Dong et al., 2022). Instead of directly predicting links, these methods perform graph classification tasks on the enclosing subgraphs sampled around the target link. These methods can achieve even better results compared to node-wise encoder-decoder frameworks by assigning node labels to indicate different roles within the subgraphs. However, constructing subgraphs poses challenges in terms of efficiency and scalability, requiring substantial computational resources. Our work focuses on the encoder-decoder framework for LP, circumventing the issues associated with subgraph construction.

**Methods for Cold-start Nodes.** Recently, several GNN-based methods (Wu et al., 2019; Liu et al., 2020; Tang et al., 2020b; Liu et al., 2021; Zheng et al., 2021) have explored degree-specific transformations to address robustness and cold-start node issues. Tang et al.(Tang et al., 2020b) introduced a degree-related graph convolutional network to mitigate degree-related bias in node classification tasks. Liu et al.(Liu et al., 2021) proposed a transferable neighborhood translation model to address missing neighbors for cold-start nodes. Zheng et al.(Zheng et al., 2021) tackled the cold-start nodes problem by recovering missing latent neighbor information. These methods require cold-start-node-specific architectural components, unlike our approach, which does not necessitate any architectural modifications. Additionally, other studies have focused on long-tail scenarios in various domains, such as cold-start recommendation(Chen et al., 2020; Lu et al., 2020; Hao et al., 2021). Imbalance tasks present another common long-tail problem, where there are long-tail instances within small classes (Lin et al., 2017; Ren et al., 2020; Tan et al., 2020; Kang et al., 2019; Tang et al., 2020a). Approaches like (Lin et al., 2017; Ren et al., 2020; Tan et al., 2020) address this issue by adapting the loss for different samples. However, due to the different problem settings, it is challenging to directly apply these methods to our tasks. We only incorporate the balanced cross entropy introduced by Lin et al. (Lin et al., 2017) as one of our baselines.

**Graph Data Augmentation.** Graph data augmentation expands the original data by perturbing or modifying the graphs to enhance the generalizability of GNNs (Zhao et al., 2022a; Ding et al., 2022). Existing methods primarily focus on semi-supervised node-level tasks(Rong et al., 2019; Feng et al., 2020; Zhao et al., 2021; Park et al., 2021) and graph-level tasks (Liu et al., 2022a; Luo et al., 2022). However, the exploration of graph data augmentation for LP remains limited (Zhao et al., 2022b). CFLP (Zhao et al., 2022b) proposes the creation of counterfactual links to learn representations from both observed and counterfactual links. Nevertheless, this method encounters scalability issues due to the high computational complexity associated with finding counterfactual links. Moreover, there exist general graph data augmentation methods (Liu et al., 2022b; Hu et al., 2022) that can be applied to various tasks. LAGNN (Liu et al., 2022b) proposed to use a generative model to provide additional neighbor features for each node. TuneUP (Hu et al., 2022) designs a two-stage training strategy, which trains GNNs twice to make them perform well on both warm nodes and cold-start nodes. These augmentation methods come with the trade-off of introducing extra runtime either before or during the model training. Unlike TLC-GNN (Yan et al., 2021), which necessitates extracting topological features for each node pair, and GIANT (Chien et al., 2021), which requires pre-training of the text encoder to improve node features, our methods are more streamlined and less complex.

## B  ADDITIONAL DATASETS DETAILS

This section provides detailed information about the datasets used in our experiments. We consider various types of networks, including citation networks, collaboration networks, and co-purchase networks. The datasets we utilize are as follows:

- Citation Networks: `Cora` and `Citeseer` originally introduced by Yang et al. (2016), consist of citation networks where the nodes represent papers and the edges represent citations between papers. `IGB-100K` (Khatua et al., 2023) is a recently-released benchmark citation network with high-quality node features and a large dataset size.

Table 5: Detailed statistics of data splits under the transductive and inductive setting.

| Datasets | Original Graph | | Testing Isolated | | Testing Low-degree | | Testing Warm | |
|---|---|---|---|---|---|---|---|---|
| **Transductive Setting** | | | | | | | | |
| | #Nodes | #Edges | #Nodes | #Edges | #Nodes | #Edges | #Nodes | #Edges |
| Cora | 2,708 | 5,278 | 135 | 164 | 541 | 726 | 662 | 1,220 |
| Citeseer | 3,327 | 4,552 | 291 | 342 | 492 | 591 | 469 | 887 |
| CS | 18,333 | 163,788 | 309 | 409 | 1,855 | 2,687 | 10,785 | 29,660 |
| Physics | 34,493 | 495,924 | 275 | 397 | 2,062 | 3,188 | 25,730 | 95,599 |
| Computers | 13,752 | 491,722 | 218 | 367 | 830 | 1,996 | 11,887 | 194,325 |
| Photos | 7,650 | 238,162 | 127 | 213 | 516 | 1,178 | 6,595 | 93,873 |
| IGB-100K | 100,000 | 547,416 | 1,556 | 1,737 | 6,750 | 7,894 | 23,949 | 35,109 |
| **Inductive Setting** | | | | | | | | |
| | #Nodes | #Edges | #Nodes | #Edges | #Nodes | #Edges | #Nodes | #Edges |
| Cora | 2,708 | 5,278 | 149 | 198 | 305 | 351 | 333 | 505 |
| Citeseer | 3,327 | 4,552 | 239 | 265 | 272 | 302 | 239 | 339 |
| CS | 18,333 | 163,788 | 1,145 | 1,867 | 1,202 | 1,476 | 6,933 | 13,033 |
| Physics | 34,493 | 495,924 | 2,363 | 5,263 | 1,403 | 1,779 | 17,881 | 42,548 |
| Computers | 13,752 | 491,722 | 1,126 | 4,938 | 239 | 302 | 9,235 | 43,928 |
| Photos | 7,650 | 238,162 | 610 | 2,375 | 169 | 212 | 5,118 | 21,225 |
| IGB-100K | 100,000 | 547,416 | 5,507 | 9,708 | 8,706 | 13,815 | 24,903 | 41,217 |

- Collaboration Networks: `CS` and `Physics` are representative collaboration networks. In these networks, the nodes correspond to authors and the edges represent collaborations between authors.

- Co-purchase Networks: `Computers` and `Photos` are co-purchase networks, where the nodes represent products and the edges indicate the co-purchase relationship between two products.

**Why there are no OGB (Hu et al., 2020) datasets applied?** OGB benchmarks that come with node features, such as OGB-collab and OGB-citation2, lack a substantial number of isolated or low-degree nodes, which makes it challenging to yield convincing results for experiments focusing on the cold-start problem. This is primarily due to the split setting adopted by OGB, where the evaluation is centered around a set of the most recent papers with high degrees. Besides, considering these datasets have their fixed splitting settings based on time, it will lead to inconsistent problems to compared with the leaderboard results if we use our own splitting method to ensure we have a reasonable number of isolated/low-degree nodes. Given these constraints, we opted for another extensive benchmark dataset, IGB-100K (Khatua et al., 2023), to test and showcase the effectiveness of our methods on large-scale graphs. We further conducted the experiments on `IGB1M`, which are shown in Appendix D.4.

## B.1 TRANSDUCTIVE SETTING

For the transductive setting, we randomly split the edges into training, validation, and testing sets based on the splitting ratio specified in Section 4.1. The nodes in training/validation/testing are all visible during the training process. However, the positive edges in validation/testing sets are masked out for training. After the split, we calculate the degrees of each node using the validation graph. The dataset statistics are shown in Table 5.

## B.2 INDUCTIVE SETTING

The inductive setting is considered a more realistic setting compared to the transductive setting, where new nodes appear after the training process. Following the inductive setting introduced in Guo et al. (2022) and Shiao et al. (2022), we perform node splitting to randomly sample 10% nodes from the original graph as the new nodes appear after the training process. The remaining nodes are considered observed nodes during the training. Next, we group the edges into three sets: observed-observed, observed-new, and new-new node pairs. We select 10% of observed-observed, 10% of observed-new, and 10% of new-new node pairs as the testing edges. We consider the remaining observed-new and

new-new node pairs, along with an additional 10% of observed-observed node pairs, as the newly visible edges for the testing inference. The datasets statistics are shown in Table 5.

## C  NODEDUP ALGORITHM

In this section, we provide a detailed description of our algorithm, which is outlined in Algorithm 1. Compared to the default training of GNNs for LP tasks, NODEDUP incorporates additional augmentation steps, denoted as L1-L5 in Algorithm 1.

---

**Algorithm 1:** NODEDUP.

---

**Require:** Graph $G = \{\mathcal{V}, \mathcal{E}, \mathbf{X}\}$, Supervision $\mathcal{Y}$, AGG, UPDATE, GNNs Layer L, DECODER, Supervised loss function $\mathcal{L}_{sup}$.
 1: **# Augment the graph by duplicating cold-start nodes $\mathcal{V}_{cold}$.**
 2: Identify cold node set $\mathcal{V}_{cold}$ based on the node degree.
 3: Duplicate all cold nodes to generate the augmented node set $\mathcal{V}' = \mathcal{V} \cup \mathcal{V}_{cold}$, whose node feature matrix is then $\mathbf{X}' \in \mathbb{R}^{(N+|\mathcal{V}_{cold}|) \times F}$.
 4: Add an edge between each cold node $v \in \mathcal{V}_{cold}$ and its duplication $v'$, then get the augmented edge set $\mathcal{E}' = \mathcal{E} \cup \{e_{vv'} : \forall v \in \mathcal{V}_{cold}\}$.
 5: Add the augmented edges into the training set and get $\mathcal{Y}' = \mathcal{Y} \cup \{y_{vv'} = 1 : \forall v \in \mathcal{V}_{cold}\}$.
 6: **# End-to-end supervised training based on the augmented graph $G' = \{\mathcal{V}', \mathcal{E}', \mathbf{X}'\}$.**
 7: **for** $l = 1$ to $L$ **do**
 8:    **for** $v$ in $\mathcal{V}'$ **do**
 9:       $\boldsymbol{h}_v'^{(l+1)} = \text{UPDATE}\left(\boldsymbol{h}_v'^{(l)}, \text{AGG}\left(\{\boldsymbol{h}_u'^{(l)}\} : \forall e_{uv} \in \mathcal{E}'\right)\right)$
10:    **end for**
11: **end for**
12: **for** $(i, j)$ in $\mathcal{Y}'$ **do**
13:    $\hat{y}_{ij}' = \sigma\left(\text{DECODER}(\boldsymbol{h}_i', \boldsymbol{h}_j')\right)$
14: **end for**
15: Loss $= \sum_{(i,j) \in \mathcal{Y}'} \mathcal{L}_{sup}(\hat{y}_{ij}', y_{ij})$

---

## D  FURTHER EXPERIMENTAL RESULTS

### D.1  SELECTION OF THE THRESHOLD $\delta$.

Our decision to set the threshold $\delta$ at 2 is grounded in data-driven analysis, as illustrated in Figure 1 and Figure 6. These figures reveal that nodes with degrees not exceeding 2 consistently perform below the average Hits@10 across all datasets, and higher than 2 will outperform the average results. Besides, our choice aligns with methodologies in previous studies (Liu et al., 2020; 2021), where cold nodes are identified using a fixed threshold across all the datasets. In addition, we conduct experiments with different thresholds $\delta$ on Cora and Citeseer datasets. The results are shown in Table 6. Our findings were consistent across different thresholds, with similar observations at $\delta = 1$, $\delta = 2$, and $\delta = 3$. This indicates that our method's effectiveness is not significantly impacted by changes in this threshold.

### D.2  PERFORMANCE ON WARM-WARM AND WARM-COLD LINKS.

To clearly explain the performance improvements of NODEDUP on Warm nodes, we first compared the number of Warm-Warm and Warm-Cold links in the testing set. Then, we conducted experiments to compare the performance of our methods on these two sets of links. The results, shown in Table 7, indicate that the number of Warm-Warm links consistently exceeds that of Warm-Cold links across all datasets. This means that Warm-Cold links do not dominate the performance of Warm nodes. Additionally, our methods consistently improve performance on Warm-Cold links while maintaining performance on Warm-Warm links. These findings demonstrate that our methods do not negatively impact the learning ability of Warm nodes. The observed improvement on Warm nodes is primarily due to better learning on Cold nodes, as we demonstrated in Section 3.2.

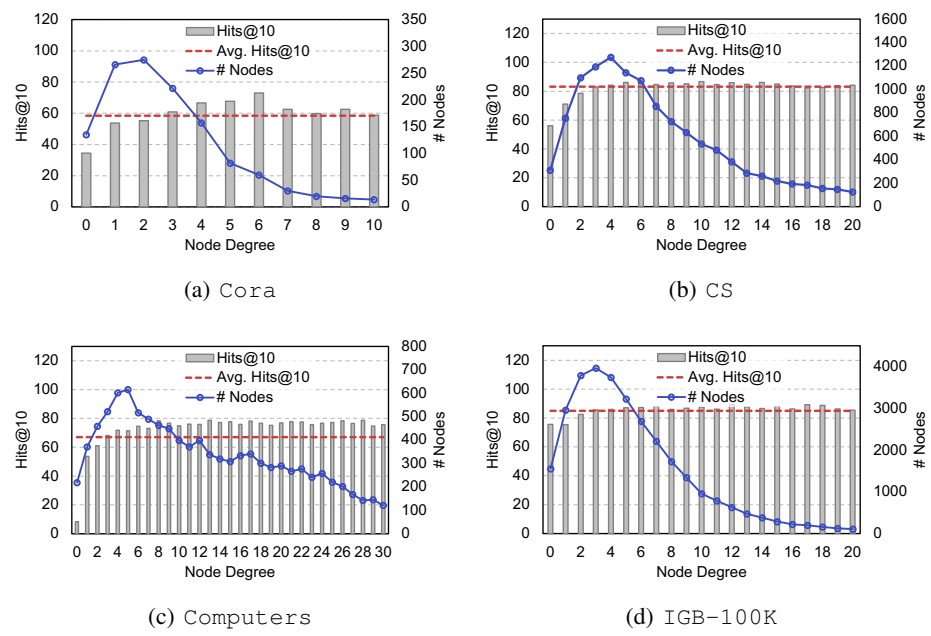

(a) Cora

(b) CS

(c) Computers

(d) IGB-100K

Figure 6: Node Degree Distribution and LP Performance Distribution w.r.t Nodes Degrees showing reverse trends on various datasets.

Table 6: Performance with different thresholds $\delta$ on Cora and Citeseer datasets.

|  |  | $\delta = 1$ |  | $\delta = 2$ |  | $\delta = 3$ |  |
|---|---|---|---|---|---|---|---|
|  |  | Gsage | NODEDUP | Gsage | NODEDUP | Gsage | NODEDUP |
| Cora | Isolated | $31.34_{\pm 5.60}$ | $42.20_{\pm 2.30}$ | $32.20_{\pm 3.58}$ | $44.27_{\pm 3.82}$ | $31.95_{\pm 1.26}$ | $43.17_{\pm 2.94}$ |
|  | Low-degree | $53.98_{\pm 1.20}$ | $57.99_{\pm 1.34}$ | $59.45_{\pm 1.09}$ | $61.98_{\pm 1.14}$ | $59.64_{\pm 1.01}$ | $62.68_{\pm 0.63}$ |
|  | Warm | $61.68_{\pm 0.29}$ | $61.17_{\pm 0.43}$ | $61.14_{\pm 0.78}$ | $59.07_{\pm 0.68}$ | $61.03_{\pm 0.79}$ | $59.91_{\pm 0.44}$ |
|  | Overall | $58.01_{\pm 0.57}$ | $59.16_{\pm 0.44}$ | $58.31_{\pm 0.68}$ | $58.92_{\pm 0.82}$ | $58.08_{\pm 0.74}$ | $59.99_{\pm 0.53}$ |
| Citeseer | Isolated | $47.25_{\pm 1.82}$ | $56.49_{\pm 1.72}$ | $47.13_{\pm 2.43}$ | $57.54_{\pm 1.04}$ | $47.31_{\pm 2.17}$ | $56.90_{\pm 1.12}$ |
|  | Low-degree | $54.10_{\pm 0.85}$ | $71.09_{\pm 0.47}$ | $61.88_{\pm 0.79}$ | $75.50_{\pm 0.39}$ | $62.97_{\pm 0.83}$ | $75.45_{\pm 0.40}$ |
|  | Warm | $72.41_{\pm 0.35}$ | $74.57_{\pm 1.04}$ | $71.45_{\pm 0.52}$ | $74.68_{\pm 0.67}$ | $73.57_{\pm 0.46}$ | $75.02_{\pm 0.84}$ |
|  | Overall | $64.27_{\pm 0.45}$ | $70.53_{\pm 0.91}$ | $63.77_{\pm 0.83}$ | $71.73_{\pm 0.47}$ | $64.05_{\pm 0.42}$ | $71.80_{\pm 0.40}$ |

Table 7: Distribution and AUC performance of testing Warm-Warm and Warm-Cold links.

|  | Warm-Warm |  |  |  | Warm-Cold |  |  |  |
|---|---|---|---|---|---|---|---|---|
|  | Number | GSage | NODEDUP(L) | NODEDUP | Number | GSage | NODEDUP(L) | NODEDUP |
| Cora | 157738 | $94.92_{\pm 0.31}$ | $95.17_{\pm 0.19}$ | $\mathbf{95.18_{\pm 0.18}}$ | 16759 | $77.06_{\pm 1.40}$ | $\mathbf{81.41_{\pm 1.18}}$ | $80.51_{\pm 1.72}$ |
| Citeseer | 63266 | $\mathbf{97.21_{\pm 0.09}}$ | $97.06_{\pm 0.21}$ | $97.02_{\pm 0.12}$ | 24020 | $85.40_{\pm 0.78}$ | $87.96_{\pm 0.79}$ | $\mathbf{88.40_{\pm 0.92}}$ |
| CS | 4209161 | $98.31_{\pm 0.03}$ | $98.30_{\pm 0.02}$ | $\mathbf{98.42_{\pm 0.02}}$ | 91458 | $87.92_{\pm 0.19}$ | $\mathbf{91.47_{\pm 0.35}}$ | $90.44_{\pm 0.84}$ |
| Physics | 11462743 | $99.01_{\pm 0.01}$ | $99.01_{\pm 0.02}$ | $\mathbf{99.02_{\pm 0.00}}$ | 103174 | $86.21_{\pm 0.33}$ | $89.94_{\pm 0.31}$ | $\mathbf{90.23_{\pm 0.51}}$ |
| Photos | 2984253 | $97.85_{\pm 0.06}$ | $\mathbf{98.03_{\pm 0.04}}$ | $97.87_{\pm 0.02}$ | 104737 | $59.80_{\pm 1.33}$ | $\mathbf{68.11_{\pm 0.43}}$ | $64.32_{\pm 0.73}$ |
| Computers | 5417165 | $97.58_{\pm 0.07}$ | $\mathbf{97.60_{\pm 0.08}}$ | $97.54_{\pm 0.09}$ | 217090 | $46.49_{\pm 0.75}$ | $57.32_{\pm 0.99}$ | $\mathbf{57.63_{\pm 0.49}}$ |
| IGB-100K | 6899924 | $98.70_{\pm 0.00}$ | $\mathbf{98.71_{\pm 0.02}}$ | $98.64_{\pm 0.01}$ | 1372994 | $97.14_{\pm 0.10}$ | $\mathbf{98.63_{\pm 0.42}}$ | $98.23_{\pm 0.06}$ |

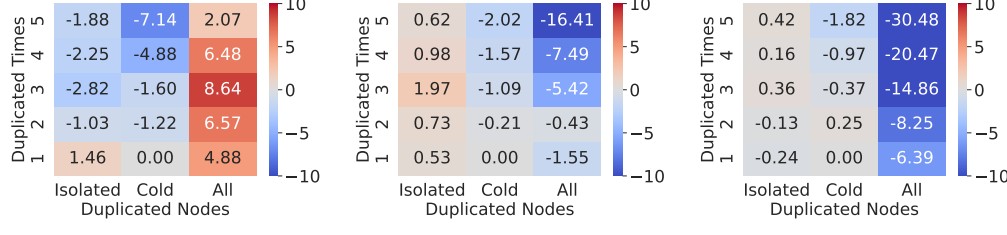

(a) Isolated          (b) Low-degree          (c) Overall

Figure 7: Ablation study on duplication frequency and nodes of NODEDUP. (a), (b), (c) show the performance for Isolated nodes, Low-degree nodes, and Overall settings, respectively. The numbers displayed in each block represent the differences compared to duplicating cold nodes once.

Table 8: Link prediction performance with different duplication nodes of NodeDup on Citeseer. "D_*" indicates duplication of "*" group nodes for one time.

|  | Isolated | Low-degree | Warm | Overall |
|---|---|---|---|---|
| Supervised | 47.13 | 61.88 | 71.45 | 63.77 |
| D_Isolated | 54.04 | 72.28 | 74.53 | 69.95 |
| D_Cold | 57.54 | 75.50 | 74.68 | 71.73 |
| D_Mid-warm | 46.93 | 61.34 | 71.84 | 63.75 |
| D_Warm | 47.49 | 62.20 | 71.54 | 63.99 |
| D_Random | 54.10 | 72.39 | 75.05 | 70.06 |
| D_All | 58.87 | 76.09 | 76.01 | 72.44 |

## D.3 INFLUENCE OF THE DUPLICATION FREQUENCY AND NODES

In our experiments, we duplicate cold nodes once and add one edge for each cold node in NODEDUP. In Figure 7, we present the results of our ablation study, focusing on the effects of duplication frequency and duplicated nodes on the performance of NODEDUP in terms of Isolated, Low-degree, and Overall settings. The numbers displayed in each block represent the differences compared to duplicating cold nodes once. We observe that increasing the duplication times does not necessarily lead to improvements across all settings, except when duplicating all nodes for Isolated nodes performance. We also notice that duplicating all nodes multiple times can significantly enhance the performance on Isolated nodes. However, this strategy negatively impacts the overall performance due to the increased number of isolated nodes in the graph. As a result, duplicating cold nodes once remains the optimal strategy, consistently yielding strong performance across all settings.

To make our analysis more comprehensive, we further conducted the experiments to show the results with duplicating warm nodes, mid-warm nodes, and randomly sampled nodes for one time, respectively, on Citeseer. The results are shown in Table 8. From the table, we can observe that duplicating mid-warm and warm nodes are not useful for the LP performance for all the settings. It's probably because for the mid-warm and warm nodes, the neighbors' information and supervised training signals are informative enough, therefore NODEDUP cannot contribute more. We can also observe that duplicating random nodes is more effective than duplicating warm nodes but less effective than duplicating cold nodes and duplicating all nodes.

## D.4 PERFORMANCE ON LARGE-SCALE DATASETS

As outlined in Section 3.1, our methods incur a minimal increase in time complexity compared to base GNNs, with the increase being linearly proportional to the number of cold nodes. This ensures scalability. Besides, the effectiveness of our method is also insensitive to dataset size. We extend our experiments to the IGB1M dataset, featuring 1 million nodes and 12 million edges. The findings, which we detail in Table 9, affirm the effectiveness of our methods in handling large-scale datasets, consistent with observations from smaller datasets.

Table 9: Performance on the large-scale dataset. The best result is **bold**. Our method consistently outperforms GSage on IGB1M.

|  |  | GSage | NodeDup |
|---|---|---|---|
| IGB1M | Isolated | $82.10_{\pm0.06}$ | $\mathbf{87.81_{\pm0.40}}$ |
|  | Low-degree | $84.73_{\pm0.06}$ | $\mathbf{90.84_{\pm0.03}}$ |
|  | Warm | $89.98_{\pm0.02}$ | $\mathbf{91.31_{\pm0.02}}$ |
|  | Overall | $89.80_{\pm0.02}$ | $\mathbf{91.29_{\pm0.02}}$ |

Table 10: Performance on heterophilic datasets. The best result for each dataset is **bold**.

| | | GSage | NodeDup(L) | NodeDup |
|---|---|---|---|---|
| Chameleon | Isolated | $24.91_{\pm6.75}$ | $\mathbf{30.76_{\pm4.02}}$ | $27.37_{\pm2.88}$ |
| | Low-degree | $79.09_{\pm1.21}$ | $80.11_{\pm0.68}$ | $\mathbf{80.91_{\pm0.41}}$ |
| | Warm | $94.00_{\pm0.23}$ | $\mathbf{94.01_{\pm0.12}}$ | $93.68_{\pm0.44}$ |
| | Overall | $92.77_{\pm0.19}$ | $\mathbf{92.88_{\pm0.10}}$ | $92.57_{\pm0.44}$ |
| Squirrel | Isolated | $25.05_{\pm3.70}$ | $\mathbf{33.07_{\pm3.20}}$ | $30.11_{\pm1.57}$ |
| | Low-degree | $63.34_{\pm2.12}$ | $66.61_{\pm0.26}$ | $\mathbf{68.05_{\pm0.80}}$ |
| | Warm | $93.35_{\pm0.22}$ | $93.43_{\pm0.11}$ | $\mathbf{93.82_{\pm0.13}}$ |
| | Overall | $92.89_{\pm0.23}$ | $93.02_{\pm0.11}$ | $\mathbf{93.41_{\pm0.13}}$ |

Table 11: Performance compared with heuristic methods and DegFairGNN (Liu et al., 2023). The best result is **bold**. NODEDUP, consistently outperforms all the heuristic methods and DegFairGNN.

| | | CN | AA | RA | DegFairGNN | GSage | NODEDUP |
|---|---|---|---|---|---|---|---|
| Cora | Isolated | 0.00 | 0.00 | 0.00 | $18.70_{\pm1.53}$ | $32.20_{\pm3.58}$ | $\mathbf{44.27_{\pm3.82}}$ |
| | Low-degree | 20.30 | 20.14 | 20.14 | $38.43_{\pm0.14}$ | $59.45_{\pm1.09}$ | $\mathbf{61.98_{\pm1.14}}$ |
| | Warm | 38.33 | 38.90 | 38.90 | $42.49_{\pm1.82}$ | $\mathbf{61.14_{\pm0.78}}$ | $59.07_{\pm0.68}$ |
| | Overall | 25.27 | 25.49 | 25.49 | $39.24_{\pm1.10}$ | $58.31_{\pm0.68}$ | $\mathbf{58.92_{\pm0.82}}$ |
| Citeseer | Isolated | 0.00 | 0.00 | 0.00 | $15.50_{\pm1.27}$ | $47.13_{\pm2.43}$ | $\mathbf{57.54_{\pm1.04}}$ |
| | Low-degree | 26.86 | 27.00 | 27.00 | $45.06_{\pm0.96}$ | $61.88_{\pm0.79}$ | $\mathbf{75.50_{\pm0.39}}$ |
| | Warm | 37.30 | 39.02 | 39.02 | $55.47_{\pm1.08}$ | $71.45_{\pm0.52}$ | $\mathbf{74.68_{\pm0.67}}$ |
| | Overall | 30.81 | 31.85 | 31.85 | $44.58_{\pm1.03}$ | $63.77_{\pm0.83}$ | $\mathbf{71.73_{\pm0.47}}$ |
| CS | Isolated | 0.00 | 0.00 | 0.00 | $17.93_{\pm1.35}$ | $56.41_{\pm1.61}$ | $\mathbf{65.87_{\pm1.70}}$ |
| | Low-degree | 39.60 | 39.60 | 39.60 | $49.83_{\pm0.68}$ | $75.95_{\pm0.25}$ | $\mathbf{81.12_{\pm0.36}}$ |
| | Warm | 72.73 | 72.74 | 72.72 | $61.72_{\pm0.37}$ | $84.37_{\pm0.46}$ | $\mathbf{84.76_{\pm0.41}}$ |
| | Overall | 69.10 | 69.11 | 69.10 | $60.20_{\pm0.37}$ | $83.33_{\pm0.42}$ | $\mathbf{84.23_{\pm0.39}}$ |
| Physics | Isolated | 0.00 | 0.00 | 0.00 | $19.48_{\pm2.94}$ | $47.41_{\pm1.38}$ | $\mathbf{66.65_{\pm0.95}}$ |
| | Low-degree | 46.08 | 46.08 | 46.08 | $47.63_{\pm0.52}$ | $79.31_{\pm0.28}$ | $\mathbf{84.04_{\pm0.22}}$ |
| | Warm | 85.48 | 85.74 | 85.70 | $62.79_{\pm0.82}$ | $90.28_{\pm0.23}$ | $\mathbf{90.33_{\pm0.05}}$ |
| | Overall | 83.87 | 84.12 | 84.09 | $62.13_{\pm0.76}$ | $89.76_{\pm0.22}$ | $\mathbf{90.03_{\pm0.05}}$ |
| Computers | Isolated | 0.00 | 0.00 | 0.00 | $9.36_{\pm1.81}$ | $9.32_{\pm1.44}$ | $\mathbf{19.62_{\pm2.63}}$ |
| | Low-degree | 28.31 | 28.31 | 28.31 | $18.90_{\pm0.81}$ | $57.91_{\pm0.97}$ | $\mathbf{61.16_{\pm0.92}}$ |
| | Warm | 59.67 | 63.50 | 62.84 | $31.44_{\pm2.25}$ | $66.87_{\pm0.47}$ | $\mathbf{68.10_{\pm0.25}}$ |
| | Overall | 59.24 | 63.03 | 62.37 | $31.27_{\pm2.22}$ | $66.67_{\pm0.47}$ | $\mathbf{67.94_{\pm0.25}}$ |
| Photos | Isolated | 0.00 | 0.00 | 0.00 | $12.99_{\pm1.51}$ | $9.25_{\pm2.31}$ | $\mathbf{17.84_{\pm3.53}}$ |
| | Low-degree | 28.44 | 28.78 | 28.78 | $20.18_{\pm0.21}$ | $52.61_{\pm0.88}$ | $\mathbf{54.13_{\pm1.58}}$ |
| | Warm | 64.53 | 67.26 | 66.88 | $42.72_{\pm0.89}$ | $67.64_{\pm0.55}$ | $\mathbf{68.68_{\pm0.49}}$ |
| | Overall | 63.94 | 66.64 | 66.26 | $42.37_{\pm0.87}$ | $67.32_{\pm0.54}$ | $\mathbf{68.39_{\pm0.48}}$ |
| IGB-100K | Isolated | 0.00 | 0.00 | 0.00 | $57.09_{\pm21.08}$ | $75.92_{\pm0.52}$ | $\mathbf{88.04_{\pm0.20}}$ |
| | Low-degree | 12.26 | 12.26 | 12.26 | $59.45_{\pm21.84}$ | $79.38_{\pm0.23}$ | $\mathbf{88.98_{\pm0.17}}$ |
| | Warm | 30.65 | 30.65 | 30.65 | $65.57_{\pm20.43}$ | $86.42_{\pm0.24}$ | $\mathbf{88.28_{\pm0.20}}$ |
| | Overall | 26.22 | 26.22 | 26.22 | $64.16_{\pm20.70}$ | $84.77_{\pm0.21}$ | $\mathbf{88.39_{\pm0.18}}$ |

## D.5 PERFORMANCE ON HETEROPHILY DATASETS

We have conducted experiments on two heterophilic datasets (i.e., Chameleon (Pei et al., 2020) and Squirrel (Pei et al., 2020)), with the results shown in Table 10. Our methods improve GNN performance across all settings on these datasets. Specifically, NodeDup and NodeDup(L) enhance the performance of Isolated nodes by 9.9% and 23.5% on Chameleon, and by 20.2% and 32.0% on Squirrel.

## D.6 PERFORMANCE COMPARED WITH HEURISTIC METHODS

We compare our method with traditional link prediction baselines, such as common neighbors (CN), Adamic-Adar(AA), Resource allocation (RA). The results are shown in Table 11. We observe that NODEDUP can consistently outperform these heuristic methods across all the datasets, with particularly significant improvements observed on Isolated nodes.

Table 12: Performance compared with base GNN model and baselines for cold-start methods (evaluated by MRR). The best result is **bold**, and the runner-up is underlined. NODEDUP and NODEDUP(L) outperform GSage and cold-start baselines almost all the cases.

| | | GSage | Imbalance | TailGNN | Cold-brew | NODEDUP(L) | NODEDUP |
|---|---|---|---|---|---|---|---|
| Cora | Isolated | $16.73_{\pm1.50}$ | $17.12_{\pm0.77}$ | $20.88_{\pm0.97}$ | $15.96_{\pm1.60}$ | $22.83_{\pm0.48}$ | $\mathbf{25.61}_{\pm1.41}$ |
| | Low-degree | $38.46_{\pm0.62}$ | $37.93_{\pm1.17}$ | $\underline{40.19}_{\pm0.96}$ | $35.20_{\pm0.55}$ | $\mathbf{40.20}_{\pm1.02}$ | $39.78_{\pm0.97}$ |
| | Warm | $\underline{36.97}_{\pm0.60}$ | $34.94_{\pm0.87}$ | $36.39_{\pm0.51}$ | $31.97_{\pm0.31}$ | $\mathbf{36.99}_{\pm0.41}$ | $35.34_{\pm0.32}$ |
| | Overall | $35.91_{\pm0.51}$ | $34.59_{\pm0.81}$ | $\underline{36.49}_{\pm0.59}$ | $31.84_{\pm0.17}$ | $\mathbf{36.89}_{\pm0.47}$ | $35.82_{\pm0.34}$ |
| Citeseer | Isolated | $29.36_{\pm2.30}$ | $28.35_{\pm1.02}$ | $22.49_{\pm1.67}$ | $21.91_{\pm5.24}$ | $\underline{34.19}_{\pm0.77}$ | $\mathbf{38.26}_{\pm1.26}$ |
| | Low-degree | $44.13_{\pm0.38}$ | $44.67_{\pm0.44}$ | $43.92_{\pm1.55}$ | $34.65_{\pm10.10}$ | $\underline{51.58}_{\pm0.56}$ | $\mathbf{53.71}_{\pm0.64}$ |
| | Warm | $46.68_{\pm0.48}$ | $46.95_{\pm1.01}$ | $45.93_{\pm1.17}$ | $36.45_{\pm7.50}$ | $\mathbf{48.89}_{\pm0.53}$ | $\underline{48.05}_{\pm0.42}$ |
| | Overall | $42.60_{\pm0.59}$ | $42.72_{\pm0.52}$ | $40.87_{\pm1.34}$ | $33.13_{\pm7.90}$ | $\underline{47.00}_{\pm0.44}$ | $\mathbf{48.05}_{\pm0.54}$ |
| CS | Isolated | $35.54_{\pm0.74}$ | $29.61_{\pm1.62}$ | $30.32_{\pm0.92}$ | $32.35_{\pm0.77}$ | $\underline{42.22}_{\pm1.41}$ | $\mathbf{44.94}_{\pm0.60}$ |
| | Low-degree | $56.18_{\pm0.81}$ | $57.44_{\pm0.68}$ | $46.66_{\pm0.61}$ | $42.67_{\pm0.26}$ | $\underline{61.20}_{\pm0.64}$ | $\mathbf{61.65}_{\pm0.84}$ |
| | Warm | $58.18_{\pm0.84}$ | $57.03_{\pm0.77}$ | $48.32_{\pm0.44}$ | $43.71_{\pm0.41}$ | $\mathbf{59.94}_{\pm0.54}$ | $\underline{58.67}_{\pm0.72}$ |
| | Overall | $57.73_{\pm0.83}$ | $56.72_{\pm0.73}$ | $47.96_{\pm0.45}$ | $43.48_{\pm0.38}$ | $\mathbf{59.83}_{\pm0.52}$ | $\underline{58.74}_{\pm0.70}$ |
| Physics | Isolated | $27.73_{\pm1.10}$ | $33.61_{\pm0.34}$ | $23.17_{\pm0.74}$ | $30.62_{\pm0.30}$ | $\underline{41.12}_{\pm1.10}$ | $\mathbf{45.62}_{\pm2.45}$ |
| | Low-degree | $61.40_{\pm0.52}$ | $62.74_{\pm0.27}$ | $47.05_{\pm0.39}$ | $41.95_{\pm0.15}$ | $\underline{64.04}_{\pm0.43}$ | $\mathbf{65.94}_{\pm0.21}$ |
| | Warm | $66.72_{\pm0.47}$ | $66.03_{\pm0.09}$ | $55.77_{\pm0.49}$ | $46.06_{\pm0.12}$ | $\mathbf{66.94}_{\pm0.49}$ | $\underline{66.83}_{\pm0.04}$ |
| | Overall | $66.39_{\pm0.47}$ | $65.80_{\pm0.09}$ | $55.36_{\pm0.49}$ | $45.86_{\pm0.12}$ | $\mathbf{66.74}_{\pm0.49}$ | $\underline{66.72}_{\pm0.04}$ |
| Computers | Isolated | $4.50_{\pm0.75}$ | $5.01_{\pm0.71}$ | $4.88_{\pm0.54}$ | $4.07_{\pm0.46}$ | $\underline{8.59}_{\pm1.45}$ | $\mathbf{9.65}_{\pm1.10}$ |
| | Low-degree | $26.65_{\pm0.62}$ | $26.85_{\pm0.31}$ | $21.22_{\pm0.56}$ | $23.40_{\pm0.59}$ | $\underline{28.85}_{\pm1.13}$ | $\mathbf{29.78}_{\pm0.32}$ |
| | Warm | $34.11_{\pm0.40}$ | $33.77_{\pm0.17}$ | $31.02_{\pm0.34}$ | $28.75_{\pm0.23}$ | $\underline{35.11}_{\pm0.31}$ | $\mathbf{35.63}_{\pm0.14}$ |
| | Overall | $33.98_{\pm0.40}$ | $33.65_{\pm0.16}$ | $30.88_{\pm0.34}$ | $28.64_{\pm0.23}$ | $\underline{35.00}_{\pm0.31}$ | $\mathbf{35.52}_{\pm0.13}$ |
| Photos | Isolated | $3.99_{\pm0.52}$ | $4.79_{\pm1.38}$ | $5.78_{\pm0.94}$ | $6.49_{\pm0.98}$ | $\mathbf{8.23}_{\pm1.10}$ | $\underline{7.90}_{\pm1.55}$ |
| | Low-degree | $25.10_{\pm1.35}$ | $24.60_{\pm1.20}$ | $20.41_{\pm1.29}$ | $21.54_{\pm0.35}$ | $\mathbf{27.90}_{\pm0.90}$ | $\underline{26.90}_{\pm1.29}$ |
| | Warm | $34.90_{\pm0.57}$ | $33.03_{\pm0.47}$ | $30.79_{\pm0.63}$ | $29.40_{\pm0.23}$ | $\mathbf{36.84}_{\pm0.55}$ | $\underline{35.69}_{\pm0.43}$ |
| | Overall | $34.71_{\pm0.57}$ | $32.87_{\pm0.47}$ | $30.60_{\pm0.63}$ | $29.26_{\pm0.22}$ | $\mathbf{36.66}_{\pm0.54}$ | $\underline{35.52}_{\pm0.43}$ |
| IGB-100K | Isolated | $53.20_{\pm0.24}$ | $50.81_{\pm0.41}$ | $45.25_{\pm0.26}$ | $48.42_{\pm0.25}$ | $\underline{59.34}_{\pm0.51}$ | $\mathbf{61.75}_{\pm0.47}$ |
| | Low-degree | $55.93_{\pm0.28}$ | $55.79_{\pm0.30}$ | $51.11_{\pm0.29}$ | $51.92_{\pm0.15}$ | $\underline{62.35}_{\pm0.49}$ | $\mathbf{63.91}_{\pm0.26}$ |
| | Warm | $\underline{61.31}_{\pm0.49}$ | $60.63_{\pm0.40}$ | $55.91_{\pm0.18}$ | $50.88_{\pm0.20}$ | $\mathbf{61.56}_{\pm0.48}$ | $61.24_{\pm0.19}$ |
| | Overall | $60.05_{\pm0.43}$ | $59.40_{\pm0.36}$ | $54.65_{\pm0.20}$ | $50.97_{\pm0.17}$ | $\underline{61.61}_{\pm0.48}$ | $\mathbf{61.73}_{\pm0.21}$ |

### D.7 MRR RESULTS COMPARED WITH THE BASE GNN MODEL AND COLD-START BASELINES

Table 12 presents the performance of our methods, evaluated using MRR, compared against the base GNN model and cold-start baselines. We can observe that NODEDUP(L) consistently achieves significant improvements over the baseline methods for Isolated and Low-degree nodes across all datasets. NODEDUP also outperforms baselines in 13 out of 14 cases on the cold nodes. This further demonstrates the superior effectiveness of our methods in addressing cold node scenarios. Furthermore, we can also observe that our methods consistently perform on par or above baseline methods in Warm nodes and the overall setting. This observation further supports the effectiveness of our methods in maintaining and even improving the performance of Warm nodes.

### D.8 EFFICIENCY COMPARISON WITH THE BASE GNN MODEL AND COLD-START BASELINES

The efficiency comparison between our methods and cold-start baselines is presented in Figure 8. We can observe that our methods and Imbalance exhibit similar efficiency, comparable to GSage. However, TailGNN and Cold-brew demand significantly more preprocessing and training time. Cold-brew, in particular, needs the most preprocessing time as it needs to train a teacher model for distillation.

### D.9 PERFORMANCE COMPARED WITH ADDITIONAL COLD-START METHODS

**Upsampling (Provost, 2000).** In Section 3, we discussed the issue of under-representation of cold nodes during the training of LP, which is the main cause of their unsatisfactory performance. To tackle this problem, one straightforward and naive approach is upsampling (Provost, 2000), which involves increasing the number of samples in the minority class. In order to further demonstrate the effectiveness of our methods, we conducted experiments where we doubled the edge sampling

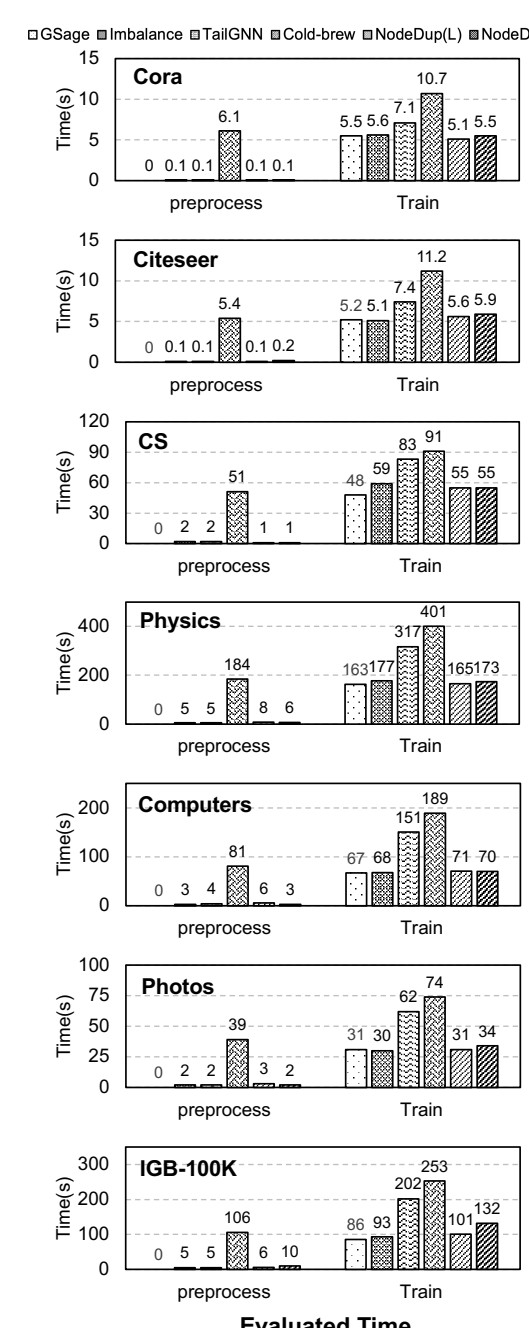

Figure 8: Time-consuming compared with cold-start methods. The histograms show the preprocessing and training time consumption of each method.

probability of code nodes, aiming to enhance their visibility. The results are presented in Table 13. We can observe that NODEDUP(L) consistently outperforms upsampling, and NODEDUP outperforms upsampling in almost all the cases, except for Warm nodes on Cora.

**The methods tackling degree bias in GNNs.** SAILOR (Liao et al., 2023) proposes a structural augmentation framework to enhance the representation learning of tail nodes. GRADE (Luo et al., 2024) improves structural fairness using graph contrastive learning methods. We used GCN as the encoder for both NODEDUP(L) and NODEDUP to ensure consistency, as both GRADE and SAILOR used GCN as their encoder. Table 14 shows that our methods outperform these baselines in all

Table 13: Performance compared with upsampling(evaluated by Hits@10). The best result is **bold**, and the runner-up is underlined. NODEDUP(L) consistently outperforms upsampling.

| Dataset | Method | Isolated | Low-degree | Warm | Overall |
|---------|--------|----------|-----------|------|---------|
| Cora | Upsampling | $32.81_{\pm2.75}$ | $59.57_{\pm0.60}$ | $60.49_{\pm0.81}$ | $57.90_{\pm0.65}$ |
| | NODEDUP(L) | $\underline{39.76}_{\pm1.32}$ | $\mathbf{62.53}_{\pm1.03}$ | $\mathbf{62.07}_{\pm0.37}$ | $\mathbf{60.49}_{\pm0.49}$ |
| | NODEDUP | $\mathbf{44.27}_{\pm3.82}$ | $\underline{61.98}_{\pm1.14}$ | $59.07_{\pm0.68}$ | $\underline{58.92}_{\pm0.82}$ |
| Citeseer | Upsampling | $46.88_{\pm0.45}$ | $62.32_{\pm1.57}$ | $71.33_{\pm1.35}$ | $63.81_{\pm0.81}$ |
| | NODEDUP(L) | $\underline{52.46}_{\pm1.16}$ | $\underline{73.71}_{\pm1.22}$ | $\mathbf{74.99}_{\pm0.37}$ | $\underline{70.34}_{\pm0.35}$ |
| | NODEDUP | $\mathbf{57.54}_{\pm1.04}$ | $\mathbf{75.50}_{\pm0.39}$ | $74.68_{\pm0.67}$ | $\mathbf{71.73}_{\pm0.47}$ |
| CS | Upsampling | $49.63_{\pm2.24}$ | $75.62_{\pm0.13}$ | $83.40_{\pm0.73}$ | $82.34_{\pm0.64}$ |
| | NODEDUP(L) | $\underline{65.18}_{\pm1.25}$ | $\mathbf{81.46}_{\pm0.57}$ | $\mathbf{85.48}_{\pm0.26}$ | $\mathbf{84.90}_{\pm0.29}$ |
| | NODEDUP | $\mathbf{65.87}_{\pm1.70}$ | $\underline{81.12}_{\pm0.36}$ | $84.76_{\pm0.41}$ | $\underline{84.23}_{\pm0.39}$ |
| Physics | Upsampling | $52.01_{\pm0.97}$ | $79.63_{\pm0.13}$ | $89.41_{\pm0.32}$ | $89.33_{\pm0.46}$ |
| | NODEDUP(L) | $\underline{65.04}_{\pm0.63}$ | $\underline{82.70}_{\pm0.22}$ | $\mathbf{90.44}_{\pm0.23}$ | $\mathbf{90.09}_{\pm0.22}$ |
| | NODEDUP | $\mathbf{66.65}_{\pm0.95}$ | $\mathbf{84.04}_{\pm0.22}$ | $90.33_{\pm0.05}$ | $\underline{90.03}_{\pm0.05}$ |
| Computers | Upsampling | $11.36_{\pm0.72}$ | $58.23_{\pm0.88}$ | $67.07_{\pm0.49}$ | $66.87_{\pm0.48}$ |
| | NODEDUP(L) | $\underline{17.11}_{\pm1.62}$ | $\mathbf{62.14}_{\pm1.06}$ | $68.02_{\pm0.41}$ | $\underline{67.86}_{\pm0.41}$ |
| | NODEDUP | $\mathbf{19.62}_{\pm2.63}$ | $\underline{61.16}_{\pm0.92}$ | $\mathbf{68.10}_{\pm0.25}$ | $\mathbf{67.94}_{\pm0.25}$ |
| Photos | Upsampling | $10.92_{\pm2.15}$ | $51.67_{\pm0.98}$ | $65.75_{\pm0.73}$ | $65.45_{\pm0.71}$ |
| | NODEDUP(L) | $\mathbf{21.50}_{\pm2.14}$ | $\mathbf{55.70}_{\pm1.38}$ | $\mathbf{69.68}_{\pm0.87}$ | $\mathbf{69.40}_{\pm0.86}$ |
| | NODEDUP | $\underline{17.84}_{\pm3.53}$ | $\underline{54.13}_{\pm1.58}$ | $68.68_{\pm0.49}$ | $68.39_{\pm0.48}$ |
| IGB-100K | Upsampling | $75.49_{\pm0.90}$ | $79.47_{\pm0.11}$ | $86.54_{\pm0.19}$ | $84.87_{\pm0.14}$ |
| | NODEDUP(L) | $\underline{87.43}_{\pm0.44}$ | $\underline{88.37}_{\pm0.24}$ | $\mathbf{88.54}_{\pm0.31}$ | $\mathbf{88.47}_{\pm0.28}$ |
| | NODEDUP | $\mathbf{88.04}_{\pm0.20}$ | $\mathbf{88.98}_{\pm0.17}$ | $\underline{88.28}_{\pm0.20}$ | $\underline{88.39}_{\pm0.18}$ |

Table 14: Performance compared with GRADE (Luo et al., 2024) and SAILOR (Liao et al., 2023). The best result is **bold**.

| | | GCN | GRADE | SAILOR | NODEDUP(L) | NODEDUP |
|---|---|-----|-------|--------|-----------|---------|
| Cora | Isolated | $40.61_{\pm3.52}$ | $43.29_{\pm2.62}$ | $45.12_{\pm1.29}$ | $42.93_{\pm2.68}$ | $\mathbf{46.71}_{\pm1.53}$ |
| | Low-degree | $63.86_{\pm0.78}$ | $58.76_{\pm1.27}$ | $62.98_{\pm3.92}$ | $\mathbf{64.63}_{\pm1.60}$ | $64.10_{\pm1.37}$ |
| | Warm | $60.59_{\pm0.62}$ | $60.00_{\pm0.51}$ | $57.34_{\pm3.80}$ | $\mathbf{61.31}_{\pm0.43}$ | $60.26_{\pm0.70}$ |
| | Overall | $60.16_{\pm0.44}$ | $56.90_{\pm0.71}$ | $58.33_{\pm3.51}$ | $\mathbf{61.02}_{\pm0.61}$ | $59.90_{\pm0.89}$ |
| Citeseer | Isolated | $45.56_{\pm1.30}$ | $50.11_{\pm2.24}$ | $49.29_{\pm2.75}$ | $47.84_{\pm0.94}$ | $\mathbf{50.64}_{\pm1.10}$ |
| | Low-degree | $69.37_{\pm0.36}$ | $59.49_{\pm1.13}$ | $65.78_{\pm1.11}$ | $70.15_{\pm1.56}$ | $\mathbf{71.13}_{\pm0.64}$ |
| | Warm | $\mathbf{74.68}_{\pm0.38}$ | $70.01_{\pm0.50}$ | $72.66_{\pm0.37}$ | $73.26_{\pm0.97}$ | $72.93_{\pm0.78}$ |
| | Overall | $67.48_{\pm0.42}$ | $61.11_{\pm0.72}$ | $64.80_{\pm0.66}$ | $67.47_{\pm0.83}$ | $\mathbf{67.67}_{\pm0.66}$ |

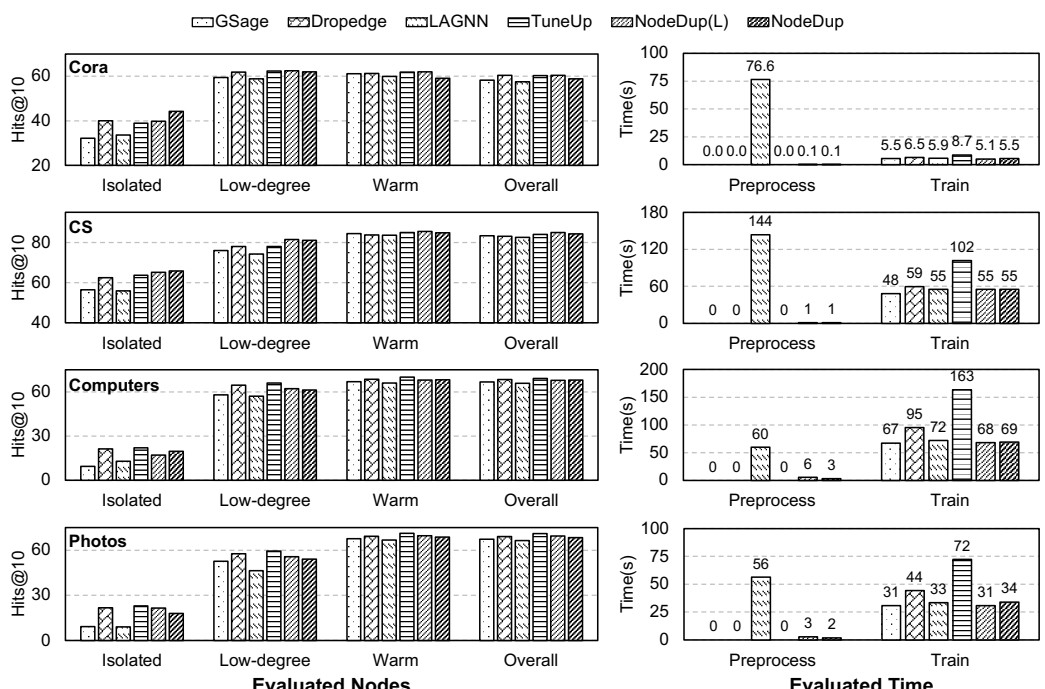

Figure 9: Performance and time-consuming compared with augmentation methods (Remaining results of Figure 5). The *left* histograms show the performance results, and the *right* histograms show the preprocessing and training time consumption of each method.

settings. Additionally, both GRADE and SAILOR perform better than vanilla GCN on Isolated nodes, which is the primary focus of their training. DegFairGNN (Liu et al., 2023) introduces a learnable debiasing function in the GNN architecture to produce fair representations for nodes, aiming for similar predictions for nodes within the same class, regardless of their degrees. Unfortunately, we've found in Table 11 that this approach is not well-suited for link prediction tasks for several reasons: (1) This method is designed specifically for node classification tasks. For example, the fairness loss, which ensures prediction distribution uniformity among low and high-degree node groups, is not suitable for link prediction because there is no defined node class in link prediction tasks. (2) This approach achieves significant performance in node classification tasks by effectively mitigating degree bias. However, in the context of link prediction, the degree trait is crucial. Applying DegFairGNN (Liu et al., 2023) would compromise the model's ability to learn from structural information, such as isomorphism and common neighbors. This, in turn, would negatively impact link prediction performance, as evidenced by references (Zhang & Chen, 2018; Chamberlain et al., 2022).

## D.10 ADDITIONAL RESULTS COMPARED WITH AUGMENTATION BASELINES

Figure 9 presents the performance compared with augmentation methods on the remaining datasets. On Cora and CS datasets, we can consistently observe that NODEDUP and NODEDUP(L) outperform all the graph augmentation baselines for Isolated and Low-degree nodes. Moreover, for Warm nodes, NODEDUP can also perform on par or above baselines. On the Computers and Photos datasets, our methods generally achieve comparable or superior performance compared to the baselines, except in comparison to TuneUP. However, it is worth noting that both NODEDUP and NODEDUP(L) exhibit more than 2× faster execution speed than TuneUP on these two datasets.

Table 15: Performance in inductive settings (Remaining results of Table 2). The best result is **bold**, and the runner-up is underlined. Our methods consistently outperform GSage.

| | | GSage | NODEDUP(L) | NODEDUP |
|---|---|---|---|---|
| Cora | Isolated | $43.64_{\pm1.84}$ | $\underline{45.31}_{\pm0.83}$ | $\mathbf{46.06}_{\pm0.66}$ |
| | Low-degree | $60.06_{\pm0.62}$ | $\underline{60.46}_{\pm0.91}$ | $\mathbf{61.94}_{\pm2.22}$ |
| | Warm | $60.59_{\pm1.13}$ | $\underline{60.95}_{\pm1.40}$ | $\mathbf{62.53}_{\pm1.23}$ |
| | Overall | $57.23_{\pm0.33}$ | $\underline{57.65}_{\pm0.82}$ | $\mathbf{59.24}_{\pm1.02}$ |
| CS | Isolated | $74.34_{\pm0.56}$ | $\underline{75.42}_{\pm0.36}$ | $\mathbf{77.80}_{\pm0.68}$ |
| | Low-degree | $75.75_{\pm0.48}$ | $\underline{77.02}_{\pm0.65}$ | $\mathbf{81.33}_{\pm0.60}$ |
| | Warm | $82.55_{\pm0.27}$ | $\underline{83.52}_{\pm0.67}$ | $\mathbf{83.55}_{\pm0.50}$ |
| | Overall | $81.00_{\pm0.28}$ | $\underline{82.01}_{\pm0.59}$ | $\mathbf{82.70}_{\pm0.52}$ |
| Computers | Isolated | $66.81_{\pm0.72}$ | $\underline{67.03}_{\pm0.51}$ | $\mathbf{69.82}_{\pm0.63}$ |
| | Low-degree | $64.17_{\pm2.01}$ | $\underline{65.10}_{\pm1.76}$ | $\mathbf{66.36}_{\pm0.69}$ |
| | Warm | $68.76_{\pm0.40}$ | $\underline{68.78}_{\pm0.39}$ | $\mathbf{70.49}_{\pm0.41}$ |
| | Overall | $68.54_{\pm0.42}$ | $\underline{68.59}_{\pm0.39}$ | $\mathbf{70.40}_{\pm0.42}$ |
| Photos | Isolated | $68.29_{\pm0.67}$ | $\underline{69.60}_{\pm0.75}$ | $\mathbf{70.46}_{\pm0.53}$ |
| | Low-degree | $63.02_{\pm1.51}$ | $\underline{64.25}_{\pm1.31}$ | $\mathbf{68.49}_{\pm2.39}$ |
| | Warm | $70.17_{\pm0.57}$ | $\underline{71.05}_{\pm0.70}$ | $\mathbf{71.61}_{\pm0.81}$ |
| | Overall | $69.92_{\pm0.57}$ | $\underline{70.84}_{\pm0.63}$ | $\mathbf{71.47}_{\pm0.77}$ |

Table 16: Performance with different encoders (Remaining results of Table 3), where the inner product is the decoder. The best result for each encoder is **bold**, and the runner-up is underlined. Our methods consistently outperform the base models, particularly for Isolated and Low-degree nodes.

| | | GAT | NODEDUP(L) | NODEDUP | JKNet | NODEDUP(L) | NODEDUP |
|---|---|---|---|---|---|---|---|
| Cora | Isolated | $25.61_{\pm1.78}$ | $\underline{30.73}_{\pm2.54}$ | $\mathbf{36.83}_{\pm1.76}$ | $30.12_{\pm1.02}$ | $\underline{37.44}_{\pm2.27}$ | $\mathbf{43.90}_{\pm3.66}$ |
| | Low-degree | $54.88_{\pm0.84}$ | $\underline{55.76}_{\pm0.50}$ | $\mathbf{56.72}_{\pm0.81}$ | $59.56_{\pm0.66}$ | $\underline{61.93}_{\pm1.64}$ | $\mathbf{62.89}_{\pm1.43}$ |
| | Warm | $\underline{55.31}_{\pm1.14}$ | $\mathbf{55.36}_{\pm1.28}$ | $53.70_{\pm1.26}$ | $\underline{58.64}_{\pm0.12}$ | $\mathbf{59.36}_{\pm1.00}$ | $57.67_{\pm1.60}$ |
| | Overall | $52.85_{\pm0.91}$ | $\mathbf{53.58}_{\pm0.80}$ | $\underline{53.43}_{\pm0.49}$ | $56.74_{\pm0.27}$ | $\mathbf{58.54}_{\pm0.83}$ | $\underline{58.40}_{\pm1.33}$ |
| CS | Isolated | $33.74_{\pm1.98}$ | $\underline{34.77}_{\pm0.90}$ | $\mathbf{41.76}_{\pm2.99}$ | $54.43_{\pm1.77}$ | $\underline{56.38}_{\pm2.14}$ | $\mathbf{64.79}_{\pm1.68}$ |
| | Low-degree | $70.20_{\pm0.47}$ | $\underline{70.90}_{\pm0.32}$ | $\mathbf{71.92}_{\pm0.36}$ | $73.97_{\pm0.72}$ | $\underline{76.64}_{\pm0.38}$ | $\mathbf{77.77}_{\pm0.43}$ |
| | Warm | $78.39_{\pm0.28}$ | $\mathbf{78.67}_{\pm0.33}$ | $77.69_{\pm0.89}$ | $82.38_{\pm0.67}$ | $\mathbf{83.29}_{\pm0.37}$ | $79.20_{\pm0.13}$ |
| | Overall | $77.16_{\pm0.24}$ | $\mathbf{77.49}_{\pm0.30}$ | $\underline{77.20}_{\pm0.80}$ | $81.35_{\pm0.62}$ | $\mathbf{82.41}_{\pm0.32}$ | $78.91_{\pm0.13}$ |
| Computers | Isolated | $12.04_{\pm2.08}$ | $\underline{16.84}_{\pm2.34}$ | $\mathbf{17.17}_{\pm2.22}$ | $9.92_{\pm3.07}$ | $\underline{23.81}_{\pm2.02}$ | $\mathbf{25.50}_{\pm1.32}$ |
| | Low-degree | $53.60_{\pm1.51}$ | $\underline{53.62}_{\pm1.00}$ | $\mathbf{53.65}_{\pm2.35}$ | $62.29_{\pm1.08}$ | $\underline{67.21}_{\pm0.99}$ | $\mathbf{68.49}_{\pm0.70}$ |
| | Warm | $\mathbf{60.19}_{\pm1.19}$ | $\underline{58.64}_{\pm0.81}$ | $58.55_{\pm1.01}$ | $69.96_{\pm0.33}$ | $\mathbf{70.90}_{\pm0.40}$ | $\underline{70.66}_{\pm0.25}$ |
| | Overall | $\mathbf{60.03}_{\pm1.19}$ | $58.50_{\pm0.80}$ | $\underline{58.77}_{\pm1.93}$ | $69.77_{\pm0.32}$ | $\mathbf{70.78}_{\pm0.40}$ | $\underline{70.55}_{\pm0.25}$ |
| Photos | Isolated | $15.31_{\pm3.46}$ | $\underline{18.03}_{\pm2.50}$ | $\mathbf{18.77}_{\pm3.33}$ | $12.77_{\pm2.40}$ | $\underline{19.44}_{\pm1.31}$ | $\mathbf{20.56}_{\pm1.61}$ |
| | Low-degree | $43.11_{\pm9.93}$ | $\underline{43.40}_{\pm9.61}$ | $\mathbf{44.21}_{\pm9.25}$ | $57.27_{\pm2.06}$ | $\underline{59.86}_{\pm1.09}$ | $\mathbf{60.93}_{\pm0.74}$ |
| | Warm | $\underline{56.17}_{\pm8.28}$ | $\mathbf{56.75}_{\pm8.33}$ | $56.10_{\pm8.35}$ | $68.35_{\pm0.81}$ | $\underline{69.56}_{\pm0.69}$ | $\mathbf{69.60}_{\pm0.50}$ |
| | Overall | $55.91_{\pm9.22}$ | $\mathbf{56.48}_{\pm8.26}$ | $\underline{55.93}_{\pm8.28}$ | $68.09_{\pm0.82}$ | $\underline{69.33}_{\pm0.68}$ | $\mathbf{69.38}_{\pm0.49}$ |

## D.11 ADDITIONAL RESULTS UNDER THE INDUCTIVE SETTING

We further evaluate and present the effectiveness of our methods under the inductive setting on the remaining datasets in Table 15. We can observe that both NODEDUP and NODEDUP(L) consistently outperform GSage for Isolated, Low-degree, and Warm nodes. Compared to NODEDUP(L), NODEDUP is particularly beneficial for this inductive setting.

## D.12 ABLATION STUDY

### D.12.1 PERFORMANCE WITH VARIOUS ENCODERS AND DECODERS

For the ablation study, we further explored various encoders and decoders on the remaining datasets. The results are shown in Table 16 and Table 17. From these two tables, we can observe that regardless of the encoders or decoders, both NODEDUP and NODEDUP(L) consistently outperform the base model for Isolated and Low-degree nodes, which further demonstrates the effectiveness of our methods on cold nodes. Furthermore, NODEDUP(L) consistently achieves better performance compared to the base model for Warm nodes.

Table 17: Link prediction performance with MLP decoder (Remaining results of Table 4), where GSage is the encoder. Our methods achieve better performance than the base model.

| | | MLP-Dec. | NODEDUP(L) | NODEDUP |
|---|---|---|---|---|
| Cora | Isolated | $16.83_{\pm2.61}$ | $\underline{37.32}_{\pm3.87}$ | $\mathbf{38.41}_{\pm1.22}$ |
| | Low-degree | $58.83_{\pm1.77}$ | $\mathbf{64.46}_{\pm2.13}$ | $\underline{64.02}_{\pm1.02}$ |
| | Warm | $\underline{58.84}_{\pm0.86}$ | $\mathbf{61.57}_{\pm0.98}$ | $58.66_{\pm0.61}$ |
| | Overall | $55.57_{\pm1.10}$ | $\mathbf{60.68}_{\pm0.66}$ | $\underline{58.93}_{\pm0.25}$ |
| CS | Isolated | $5.60_{\pm1.14}$ | $\underline{58.68}_{\pm0.95}$ | $\mathbf{60.20}_{\pm0.68}$ |
| | Low-degree | $71.46_{\pm1.08}$ | $\underline{78.82}_{\pm0.68}$ | $\mathbf{79.58}_{\pm0.31}$ |
| | Warm | $84.54_{\pm0.32}$ | $\mathbf{85.88}_{\pm0.22}$ | $\underline{85.20}_{\pm0.24}$ |
| | Overall | $82.48_{\pm0.32}$ | $\mathbf{84.96}_{\pm0.25}$ | $\underline{84.42}_{\pm0.22}$ |
| Computers | Isolated | $6.13_{\pm3.63}$ | $\mathbf{27.74}_{\pm3.38}$ | $\underline{26.70}_{\pm3.98}$ |
| | Low-degree | $62.56_{\pm1.34}$ | $\underline{62.60}_{\pm3.38}$ | $\mathbf{63.35}_{\pm3.64}$ |
| | Warm | $69.72_{\pm1.31}$ | $\mathbf{70.01}_{\pm2.41}$ | $68.43_{\pm2.50}$ |
| | Overall | $69.53_{\pm1.30}$ | $\mathbf{69.91}_{\pm3.11}$ | $68.30_{\pm2.51}$ |
| Photos | Isolated | $6.34_{\pm2.67}$ | $\underline{18.15}_{\pm2.02}$ | $\mathbf{18.97}_{\pm1.71}$ |
| | Low-degree | $55.63_{\pm6.21}$ | $\mathbf{56.13}_{\pm6.36}$ | $\underline{55.93}_{\pm7.27}$ |
| | Warm | $\underline{70.40}_{\pm6.84}$ | $\mathbf{70.67}_{\pm6.30}$ | $69.97_{\pm5.07}$ |
| | Overall | $\underline{69.89}_{\pm6.81}$ | $\mathbf{69.93}_{\pm6.24}$ | $69.69_{\pm5.07}$ |

Table 18: Performance with GCN (Kipf & Welling, 2016a) and GT (Dwivedi & Bresson, 2020) encoders, where the inner product is the decoder. The best result for each encoder is **bold**.

| | | GCN | GCN+NodeDup(L) | GCN+NodeDup | GT | GT+NodeDup(L) | GT+NodeDup |
|---|---|---|---|---|---|---|---|
| Cora | Isolated | $40.61_{\pm3.52}$ | $42.93_{\pm2.68}$ | $\mathbf{46.71}_{\pm1.53}$ | $20.93_{\pm2.46}$ | $\mathbf{38.82}_{\pm1.27}$ | $37.40_{\pm1.53}$ |
| | Low-degree | $63.86_{\pm0.78}$ | $\mathbf{64.63}_{\pm1.60}$ | $64.10_{\pm1.37}$ | $58.59_{\pm0.29}$ | $61.16_{\pm1.08}$ | $\mathbf{61.39}_{\pm0.89}$ |
| | Warm | $60.59_{\pm0.62}$ | $\mathbf{61.31}_{\pm0.43}$ | $60.26_{\pm0.70}$ | $58.14_{\pm1.15}$ | $\mathbf{59.29}_{\pm0.84}$ | $59.07_{\pm0.05}$ |
| | Overall | $60.16_{\pm0.44}$ | $\mathbf{61.02}_{\pm0.61}$ | $59.90_{\pm0.89}$ | $55.40_{\pm0.43}$ | $\mathbf{58.34}_{\pm0.19}$ | $58.18_{\pm0.42}$ |
| Citeseer | Isolated | $45.56_{\pm1.30}$ | $47.84_{\pm0.94}$ | $\mathbf{50.64}_{\pm1.10}$ | $36.84_{\pm3.26}$ | $51.46_{\pm1.27}$ | $\mathbf{52.34}_{\pm1.46}$ |
| | Low-degree | $69.37_{\pm0.36}$ | $70.15_{\pm1.56}$ | $\mathbf{71.13}_{\pm0.64}$ | $60.24_{\pm1.18}$ | $72.98_{\pm1.54}$ | $\mathbf{73.77}_{\pm1.03}$ |
| | Warm | $\mathbf{74.68}_{\pm0.38}$ | $73.26_{\pm0.97}$ | $72.93_{\pm0.78}$ | $71.14_{\pm1.47}$ | $74.48_{\pm1.08}$ | $\mathbf{75.08}_{\pm0.63}$ |
| | Overall | $67.48_{\pm0.42}$ | $67.47_{\pm0.83}$ | $\mathbf{67.67}_{\pm0.66}$ | $61.15_{\pm1.57}$ | $69.67_{\pm1.10}$ | $\mathbf{70.38}_{\pm0.86}$ |

Besides GSage, GAT and JKNet, we also conducted further experiments with convolutional-based GNNs, such as GCN (Kipf & Welling, 2016a) and GT(GraphTransformer) (Dwivedi & Bresson, 2020). The results are shown in Table 18. Our findings indicate that our methods can also improve performance when using GCN and GT as the encoder. However, since GCN uses the same matrix for both self-representations and neighbor representations, our methods only benefit from the supervision aspect. This leads to less pronounced performance improvements on cold nodes compared to using GT and GSage as the encoder. Specifically, NodeDup shows a 13.10% improvement for GCN, 60.38% for GT, and 29.79% for GSage on isolated nodes. Moverover, NodeDup(L) on average improves GCN by 5.4%, GT by 62.58%, and GSage by 17.4%.

### D.12.2 PERFORMANCE WITH SEAL (ZHANG & CHEN, 2018)

Considering our methods are flexible to integrate with GNN-based link prediction structures, we conduct the experiments on top of SEAL (Zhang & Chen, 2018) on the `Cora` and `Citeseer` datasets. The results are shown in Table 19. We can observe that adding NODEDUP on top of SEAL can consistently improve link prediction performance in the Isolated and Low-degree node settings on these two datasets.

Table 19: Performance with SEAL (Zhang & Chen, 2018) on `Cora` and `Citeseer` datasets.

| | | SEAL | SEAL + NODEDUP |
|---|---|---|---|
| Cora | Isolated | $62.20_{\pm1.06}$ | $\mathbf{70.73_{\pm0.61}}$ |
| | Low-degree | $66.80_{\pm2.83}$ | $\mathbf{67.70_{\pm4.11}}$ |
| | Warm | $\mathbf{56.69_{\pm2.36}}$ | $54.87_{\pm1.61}$ |
| | Overall | $60.60_{\pm2.38}$ | $\mathbf{60.89_{\pm2.36}}$ |
| Citeseer | Isolated | $56.92_{\pm5.53}$ | $\mathbf{66.37_{\pm1.01}}$ |
| | Low-degree | $64.13_{\pm2.56}$ | $\mathbf{65.54_{\pm1.69}}$ |
| | Warm | $58.81_{\pm3.22}$ | $\mathbf{60.73_{\pm2.75}}$ |
| | Overall | $60.18_{\pm2.98}$ | $\mathbf{63.35_{\pm1.43}}$ |

## E  IMPLEMENTATION DETAILS

In this section, we introduce the implementation details of our experiments. Our implementation can be found at `https://anonymous.4open.science/r/NodeDup-0241/README.md`.

**Parameter Settings.** We use 2-layer GNN architectures with 256 hidden dimensions for all GNNs and datasets. The dropout rate is set as 0.5. We report the results over 10 random seeds. Hyperparameters were tuned using an early stopping strategy based on performance on the validation set. We manually tune the learning rate for the final results. For the results with the inner product as the decoder, we tune the learning rate over range: $lr \in \{0.001, 0.0005, 0.0001, 0.00005\}$. For the results with MLP as the decoder, we tune the learning rate over range: $lr \in \{0.01, 0.005, 0.001, 0.0005\}$.

**Hardware and Software Configuration** All methods were implemented in Python 3.10.9 with Pytorch 1.13.1 and PyTorch Geometric (Fey & Lenssen, 2019). The experiments were all conducted on an NVIDIA P100 GPU with 16GB memory.

## F  LIMITATIONS

In our work, NODEDUP and NODEDUP(L) are specifically proposed for LP tasks. Although cold-start is a widespread issue in all graph learning tasks, our proposed methods might not be able to generalize to other tasks, such as node classification, due to their unique design. Furthermore, the two heterophily datasets we used for evaluation involve graphs where nodes with similar features are assigned different labels. Our methods may struggle on heterophilic graphs where connected nodes have dissimilar features, such as molecular networks, which are beyond the scope of this study.

