# OpenReview forum: "Node Duplication Improves Cold-start Link Prediction"
_ICLR.cc/2025/Conference — Submitted to ICLR 2025_

### Official Review · Reviewer_Abvk · 2024-10-29

**Soundness:** 2
**Presentation:** 3
**Contribution:** 1
**Rating:** 3
**Confidence:** 4

**Summary:**

This paper introduces NODEDUP, a data augmentation method designed to enhance the performance of GNNs on low-degree nodes in downstream link prediction tasks. By duplicating low-degree nodes and creating links between nodes and their own duplicates, NODEDUP can significantly improve prediction performance on *cold nodes* without compromising overall performance on *warm nodes*. Extensive experimental results demonstrate the effectiveness of NODEDUP compared to base GNNs, cold-start GNNs, and other data augmentation GNNs. Additionally, NODEDUP is a plug-and-play module that can be easily applied to different GNNs with minimal additional cost.

**Strengths:**

S1: This paper focuses on addressing the cold-start problem in link prediction tasks, a significant issue with numerous practical applications in real-world scenarios.

S2: The authors conduct extensive experiments to demonstrate the effectiveness of their method compared to traditional GNNs, cold-start GNNs, and other data augmentation techniques.

S3: The proposed method is simple and can be easily applied to different GNNs.

**Weaknesses:**

**W1**: The primary concern of this paper is its limited novelty. The main idea of duplicating cold nodes is not a novel strategy within existing GNN frameworks. Additionally, the distinction between NodeDup and the common data pre-processing step of adding self-loops is not clear. When adopting general message passing GNNs, isolated nodes can also treat themselves as neighbors and update node representations using both $\boldsymbol{W}\_1$ and $\boldsymbol{W}\_2$. Without a clear differentiation, it is challenging to justify the novelty of NodeDup over existing techniques.

**W2**: It is unclear why such a simple strategy significantly enhances overall GNN performance on warm nodes, particularly on the *citeseer* dataset, which contains a large number of isolated nodes with zero degrees. Consider an example in online social networks, where a node $u$ represents a famous individual with millions of followers, and most of $u$’s followers may have few connections with other users, serving as *cold nodes*. In this scenario, adding self-loops using NodeDup(L) on these cold nodes may introduce additional noise in predicting the *warm node* $u$. This could potentially degrade the model's performance rather than improve it.

**W3**: Additional theoretical analysis from a spectral perspective, such as examining changes in the graph Laplacian after self-augmentation, could enhance the understanding of the self-augmentation strategy. Such analysis might provide deeper insights into why and how the self-augmentation strategy works, potentially revealing underlying mechanisms that contribute to its effectiveness or limitations.

**W4**: More experiments on larger datasets, such as collab, ppa, and citation2 from the OGB benchmark datasets [R1], should be included in the main text. The current experimental setup may not be sufficient to generalize the findings across different types of graphs and scales. Including these larger datasets would provide a more comprehensive evaluation of the method scalability and robustness. Additionally, a comparison between NODEDUP and more recent link prediction models, such as MPLP [R2], is necessary.

---

[R1] W. Hu, M. Fey, M. Zitnik, Y. Dong, H. Ren, B. Liu, M. Catasta and J. Leskovec. Open Graph Benchmark: Datasets for Machine Learning on Graphs. 2020. NeurIPS(33): 22118-22133.

[R2] K. Dong, Z. Guo, N. V. Chawla. Pure Message Passing Can Estimate Common Neighbor for Link Prediction. arXiv:2309.00976.

**Questions:**

**Q1**: What is the key insight and contribution of the proposed heuristic strategy?

**Q2**: Figure 2 shows that NodeDup significantly improves performance compared to GraphSage on cold nodes while achieving results comparable to GraphSage on warm nodes. However, it raises the question of why the overall performances of two models are nearly identical.

**Q3**: The additional time complexity of the decoder should be $O( |\mathcal{V}\_{cold}| D)$ rather than $O((M+|\mathcal{V}\_{cold}|) D)$?

**Q4**: Whether self-loops are included in the adjacency matrix of the datasets used or not?

**Q5**: Further explanation about the differences in performance between NodeDup and NodeDup(L) on warm nodes is needed.

**Q6**: Additionally, there are minor typos, such as in line 400, where "did" is incorrectly used.

---

### Official Review · Reviewer_Ze8q · 2024-11-02

**Soundness:** 1
**Presentation:** 2
**Contribution:** 2
**Rating:** 3
**Confidence:** 4

**Summary:**

In this paper, the authors propose NODEDUP, a simple yet effective augmentation technique aimed at improving link prediction (LP) performance for low-degree nodes in Graph Neural Networks (GNNs) while preserving performance for high-degree nodes. NODEDUP works by duplicating low-degree nodes and creating links to their duplicates before applying standard supervised LP training. This “multi-view” approach significantly enhances LP performance for low-degree nodes without negatively impacting high-degree nodes. As a plug-and-play module, NODEDUP integrates easily into existing GNNs with minimal computational cost. Extensive experiments show average relative improvements of 38.49%, 13.34%, and 6.76% for isolated, low-degree, and warm nodes, respectively, compared to traditional GNNs and state-of-the-art cold-start methods.

**Strengths:**

1. The paper is clearly written and easy to understand.

2. The authors claim that the proposed method demonstrates significant improvements in performance.

**Weaknesses:**

1. The problem addressed in this paper does not align with the experimental datasets used. The cold start issue is a significant challenge in recommendation systems, and while the authors frequently mention the intent to tackle this problem, they rely on publicly available general graph datasets, such as citation networks, for their experiments. Crucially, they do not utilize any recommendation system datasets, such as Movielens-1M or Amazon-Books.

2. The work lacks theoretical guarantees. Although the authors attempt to explain in Section 3.2 how node duplication aids cold start link prediction, I would prefer to see definitive theorems or lemmas presented in the paper. This would provide a more solid theoretical foundation for the work.

3. In Table 18, GCN+NodeDup shows a decline in overall metrics compared to GCN. What is the reason for this phenomenon?

4. There are several works designed to solve link prediction problems using GNN methods, such as references [1], [2], and [3]. However, the authors do not compare NodeDup against these methods in their experiments.

5. The method description lacks clarity. In the NodeDup approach, determining which nodes are cold starts is a crucial step. However, in the core algorithm (Algorithm 1), the authors do not demonstrate how to deterministically obtain the set of cold start nodes V_{cold}.

6. The authors should provide a direct comparison with adding self-loops as an augmentation baseline is missing, which could help clarify the advantages of NodeDup and NodeDup(L) over simpler alternatives.

7. The authors state that OGB datasets were not used because they ``lack a substantial number of isolated or low-degree nodes'' (Lines 892-893). However, even though OGB datasets may primarily include high-degree nodes, they still contain a subset of lower-degree nodes, which would allow for a relevant evaluation of the proposed method's performance across both node types. Additionally, testing on these datasets could strengthen the claim that the method does not compromise warm node performance, as OGB is a widely recognized benchmark.

[1] Bai Q, et al. HGWaveNet: A Hyperbolic Graph Neural Network for Temporal Link Prediction. WWW-23.

[2] Zhu Z, et al. Neural Bellman-Ford Networks: A General Graph Neural Network Framework for Link Prediction. NeurIPS-21.

[3] Cai L, et al. Line Graph Neural Networks for Link Prediction. TPAMI-21.

**Questions:**

Please see my previous comment.

---

### Official Review · Reviewer_wxG5 · 2024-11-04

**Soundness:** 2
**Presentation:** 3
**Contribution:** 2
**Rating:** 3
**Confidence:** 4

**Summary:**

The paper proposes a simple yet effective graph data augmentation method called NodeDup to improve the performance of Graph Neural Networks (GNNs) in link prediction tasks, particularly for low-degree or "cold" nodes. The method duplicates low-degree nodes and creates links to their duplicates, thereby providing a "multi-view" perspective during training. It addresses the well-known cold-start problem in recommendation systems and other applications by augmenting the training data without sacrificing the performance of well-connected nodes.

**Strengths:**

The proposed method stands out for its simplicity and ease of integration with existing GNN architectures. By requiring only the duplication of low-degree nodes and the establishment of connections between these duplicates, it provides a straightforward augmentation technique that can be seamlessly applied to various GNN models.

The paper has extensive experiments to demonstrate the effectiveness and superiority of the proposed method. By evaluating the approach across multiple benchmark datasets, the authors provide strong empirical evidence supporting their claims.

**Weaknesses:**

The paper appears to lack sufficient novelty in its contributions to the field of link prediction. The underlying idea of augmenting data through duplication has been explored in various contexts. Additionally, the paper does not convincingly demonstrate how NodeDup outperforms or fundamentally enhances prior methods, leading to concerns about the overall impact of the contribution.

Although the paper provides some justification for the proposed method, it could delve deeper into the theoretical foundations of why node duplication is particularly effective for cold-start link prediction. More comprehensive discussions could help solidify the rationale behind the approach.

The paper lacks a thorough investigation into the node degree distribution of the graphs used in the experiments, which can significantly impact the efficiency of the proposed method. In cases where the graph exhibits a highly skewed degree distribution, the additional complexity introduced by duplicating cold-start nodes could potentially lead to a doubling of the original computational complexity.

**Questions:**

In Section 3.2, you mention that duplicating cold nodes provides additional view for aggregation. However, could you clarify how this extra view differs fundamentally from simply adjusting the weights assigned to a node's own representation? What specific advantages does node duplication offer that cannot be achieved through weight adjustments alone? And how is it related to helping cold-start LP

In Section 3.2, you argue that more supervision signals for cold nodes can lead to better-quality embeddings. Can you provide a more detailed explanation of how the link between a cold node and its duplicate functions as a meaningful supervision signal? What evidence or reasoning supports the claim that this relationship enhances the learning process in a way that traditional training methods (e.g. self-loop) do not?

In your paper, you define cold-start nodes using a fixed threshold of 2 for node degrees. Given that the effectiveness of this threshold may vary based on different graph properties, such as node degree distribution, and overall connectivity, how do you justify the choice of this specific threshold? Have you considered the impact of varying this threshold on the identification of cold-start nodes and the subsequent performance of your proposed method?

Is there any reasoning why NodeDup and NodeDup(L) consistently outperforms GSage in the warm nodes and overall settings. As explained in the paper, the data augmentation is done on cold-start nodes only. How does it affect the performance of the warm nodes and overall settings?

---

### Official Review · Reviewer_oyer · 2024-11-04

**Soundness:** 3
**Presentation:** 3
**Contribution:** 2
**Rating:** 5
**Confidence:** 3

**Summary:**

This paper introduces NODEDUP, a novel augmentation technique aimed at enhancing the performance of graph neural networks (GNNs) in link prediction (LP) tasks, specifically for cold-start or low-degree nodes. By duplicating low-degree nodes and linking each to its duplicate, NODEDUP provides a “multi-view” representation that improves embeddings for cold nodes while retaining the performance of high-degree nodes. Extensive experimental results demonstrate NODEDUP's ability to achieve significant LP performance gains on cold nodes across multiple datasets, highlighting its potential to address limitations in existing GNN-based LP methods.

**Strengths:**

1. **Comprehensive Experimental Evaluation**:

   The paper is well-supported by extensive experimental results, covering a wide range of datasets and comparisons. This thorough empirical analysis demonstrates NODEDUP’s efficacy in addressing the cold-start problem, adding substantial credibility and depth to the findings.

2. **Clear Presentation and Visualization**:

   The paper is well-organized and clearly written, with visual aids that effectively communicate the experimental results. The figures and tables are particularly helpful in understanding NODEDUP’s impact across different settings, making the paper accessible and easy to follow.

**Weaknesses:**

While NODEDUP demonstrates effectiveness in addressing the cold-start link prediction problem, several critical limitations should be considered:

1. **Limited Scalability**:

   NODEDUP's design focuses heavily on the "duplication of cold nodes," making it overly specialized for the cold-start problem and potentially too simple for broader link prediction tasks. The technique lacks flexibility to handle other challenges, such as heterophilic graphs or highly dynamic networks, which require adaptable augmentation strategies. This limits NODEDUP’s versatility as a universal approach in graph contrastive learning.

2. **Lack of Theoretical Foundation**:

   Despite extensive empirical results, the paper does not provide a theoretical explanation for why NODEDUP improves cold-node representation. A theoretical analysis would strengthen the validity and interpretability of the method, clarifying how and why node duplication facilitates performance gains for cold nodes in GNNs.

3. **Outdated Baselines**:

   Although the authors claim that NODEDUP does not compromise overall performance while addressing the cold-start problem, they compare it only against older baselines rather than the latest advancements in graph contrastive learning. This raises questions about how NODEDUP’s performance aligns with state-of-the-art methods and whether it may lag behind the latest graph contrastive learning approaches in overall link prediction performance.

4. **High Dependency on Hyperparameters**:

   The proposed NODEDUP method relies heavily on the selection of the hyperparameter δ, which determines the distinction between cold and warm nodes. Although the authors provide a heuristic method and a hyperparameter search experiment, they do not offer a clear rationale for selecting a specific value of δ. This raises concerns that an optimal δ may need to be tuned individually for each dataset, potentially limiting the method’s practical applicability and generalizability.

**Questions:**

1. Could the authors provide a theoretical explanation or analysis to clarify why NODEDUP’s node duplication approach is effective for improving cold-node representation?
2. Since the authors claim that their method does not compromise warm-node performance, could they provide comparisons with some of the latest graph contrastive learning baselines, such as GCLMI [1], Sp^2GCL [2], POT [3], and GraphACL [4], proposed in 2024?



[1] Xu, Yuhua, et al. "Graph contrastive learning with min-max mutual information." *Information Sciences* 665 (2024): 120378.

[2] Bo, Deyu, et al. "Graph contrastive learning with stable and scalable spectral encoding." *Advances in Neural Information Processing Systems* 36 (2024).

[3] Yu, Yue, et al. "Provable training for graph contrastive learning." *Advances in Neural Information Processing Systems* 36 (2024).

[4] Xiao, Teng, et al. "Simple and asymmetric graph contrastive learning without augmentations." *Advances in Neural Information Processing Systems* 36 (2024).

---

### Meta-Review · Area_Chair_ghew · 2024-12-06

**Metareview:**

The paper presents a data augmentation technique to enhance the accuracy of link prediction in cold-start scenarios. The method involves duplicating low-degree nodes and creating links to their duplicates prior to applying standard supervised link prediction training, thereby offering a "multi-view" perspective during training. This approach addresses the well-known cold-start problem in recommendation systems and other applications by augmenting the training data without compromising the performance of well-connected nodes.


The reviewers are unanimous in their decision that the paper is not ready for publication in its current form. The concerns raised include lack of proper benchmarks, theoretical grounding, and novelty. The authors did not post a rebuttal and hence the concerns remain unaddressed.

**Additional Comments On Reviewer Discussion:**

The authors did not post a rebuttal. Hence, there was no follow-up post-rebuttal discussion.

---

### Decision · Program_Chairs · 2025-01-22

Reject